# ImageBrush: Learning Visual In-Context Instructions for Exemplar-Based Image Manipulation

**Yasheng Sun**[*]
Tokyo Institute of Technology
sun.y.aj@m.titech.ac.jp

**Yifan Yang**[*]
Microsoft
yifanyang@microsoft.com

**Houwen Peng**
Microsoft
houwen.peng@microsoft.com

**Yifei Shen**
Microsoft
yifeishen@microsoft.com

**Yuqing Yang**
Microsoft
yuqing.yang@microsoft.com

**Han Hu**
Microsoft
hanhu@microsoft.com

**Lili Qiu**
Microsoft
liliqiu@microsoft.com

**Hideki Koike**
Tokyo Institute of Technology
koike@c.titech.ac.jp

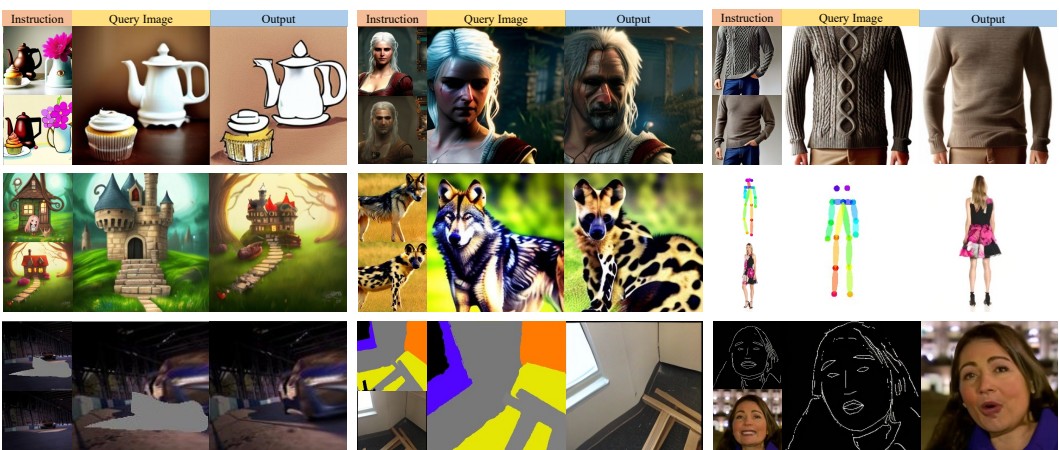

Figure 1: Demo results of the proposed **ImageBrush** framework on various image manipulation tasks. By providing a pair of *task-specific examples and a new query image* that share a similar context, ImageBrush accurately *identifies the underlying task* and generates the desired output.

## Abstract

While language-guided image manipulation has made remarkable progress, the challenge of how to instruct the manipulation process faithfully reflecting human intentions persists. An accurate and comprehensive description of a manipulation task using natural language is laborious and sometimes even impossible, primarily due to the inherent uncertainty and ambiguity present in linguistic expressions. Is it feasible to accomplish image manipulation without resorting to external cross-modal language information? If this possibility exists, the inherent modality gap would be effortlessly eliminated. In this paper, we propose a novel manipulation methodology, dubbed *ImageBrush*, that learns visual instructions for more accurate image editing. Our key idea is to employ *a pair of transformation images* as *visual instructions*, which not only precisely captures human intention but also facilitates accessibility in real-world scenarios. Capturing visual instructions is particularly challenging because it involves extracting the underlying intentions solely from visual demonstrations and then applying this operation to a new image. To address this challenge, we formulate visual instruction learning as a diffusion-based inpaint-

---

[*]Equal contribution

37th Conference on Neural Information Processing Systems (NeurIPS 2023).

ing problem, where the contextual information is fully exploited through an iterative process of generation. A visual prompting encoder is carefully devised to enhance the model's capacity in uncovering human intent behind the visual instructions. Extensive experiments show that our method generates engaging manipulation results conforming to the transformations entailed in demonstrations. Moreover, our model exhibits robust generalization capabilities on various downstream tasks such as pose transfer, image translation and video inpainting.

## 1 Introduction

Image manipulation has experienced a remarkable transformation in recent years [48, 14, 35, 71, 21, 29] . The pursuit of instructing manipulation process to align with human intent has garnered significant attention. Language, as a fundamental element of human communication, is extensively studied to guide the manipulation towards intended direction [16, 34, 6, 55, 33, 58, 5]. Despite the universal nature of language-based instructions, there are cases where they fall short in expressing certain world concepts. This limitation necessitates additional efforts and experience in magic prompt engineering, as highlighted in [64]. To compensate linguistic ambiguity, some studies [65, 32, 66] attempt to introduce visual guidance to the manipulation process. However, these approaches heavily rely on cross-modal alignment, which may not always be perfectly matched, thereby resulting in limited adaptability to user instructions.

*Is it conceivable to accomplish image manipulation exclusively through visual instructions?* If such a capability were attainable, it would not only mitigate the aforementioned cross-modality disparity but also introduce a novel form of interaction for image manipulation. Taking inspiration from the exceptional in-context capabilities by large language models [37, 40, 7] like ChatGPT and GPT-4, we propose to conceptualize the image manipulation task as a visual prompting process. This approach utilizes paired examples to establish visual context, while employing the target image as a query. Recent works [61, 4] reveal the visual in-context ability through a simple Mask Image Modeling [15] scheme. But these researches focus on understanding tasks such as detection and segmentation while only very limited context tasks are studied. Therefore, the in-context potential for image manipulation, where numerous editing choices are involved, is still a promising avenue for exploration.

In contrast to language, which primarily communicates abstract concepts, exemplar-based visual instruction explicitly concretizes manipulation operations within the visual domain. In this way, the network could directly leverage their textures, which eases the hallucination difficulty for some inexpressible concepts (*e.g.*, artist style or convoluted object appearance). On the other hand, the pairwise cases directly exhibit transformation relationship from the visual perspective, thereby facilitating better semantic correspondence learning. In this paper, we propose an *Exemplar-Based Image Manipulation* framework, **ImageBrush**, which achieves adaptive image manipulation under the instruction of a pair of exemplar demonstrations. This paradigm holds promising potential for various innovative applications. For instance, photographers can seamlessly apply Photoshop-style retouching to entire collections of similar images, enhancing their workflow. Additionally, users who encounter compelling editing instances from peers or pre-trained models can effortlessly apply these modifications to their own images, eliminating the need for original intricate intermediate steps or specific parameter adjustments.

The key is to devise *a generative model that tackles both pairwise visual instruction understanding and image synthesis* in an unified manner. To establish the intra-correlations among exemplar transformations and their inter-correlation with the query image, we utilize a grid-like image as model input that concatenates a manipulation example and a target query as [4, 61, 17]. The in-context image manipulation is accomplished by inpainting the answer picture. Unlike their approaches forwarding once for final result, we adopt the diffusion process to iteratively enhance in-context learning and refine the synthesized image. Such practice not only provide stable training objectives [32], but also mimics the behavior of a painter [9] who progressively fills and tweaks the details. Although the aforementioned formulation is capable of handling correlation modeling and image generation in a single stroke, it places a significant computational burden on the network, due to the intricate concepts and processes involved. To address this challenge, we delicately design a visual prompting encoder that maximizes the utilization of contextual information, thereby alleviating the complexity of learning. Specifically, we extract the features of each image and further process them through a transformer module for effective feature exchanging and integration. The obtained features are then injected to a diffusion network with cross-attention to augment its understanding capability. In line

with the SAM concept [27], we have also introduced an optional interface design module aimed at further improving human intent comprehension.

Our main contributions can be summarized as follows: **1**) We introduce a novel image manipulation protocol that enables the accomplishment of numerous operations through an in-context approach. **2**) We delicately devise a diffusion-based generative framework coupled with a hybrid context injection strategy to facilitate better correlation reasoning. **3**) Extensive experiments demonstrate our approach generates compelling manipulation results aligned with human intent and exhibits robust generalization abilities on various downstream tasks, paving the way for future vision foundation models.

## 2    Related Work

**Language-Guided Image Manipulation.**    The domain of generating images from textual input [45, 47, 50] has experienced extraordinary advancements, primarily driven by the powerful architecture of diffusion models [19, 47, 51, 52]. Given rich generative priors in text-guided models, numerous studies [33, 11, 36, 2, 2] have proposed their adaptation for image manipulation tasks. To guide the editing process towards the intended direction, researchers have employed CLIP [44] to fine-tune diffusion models. Although these methods demonstrate impressive performance, they often require expensive fine-tuning procedures [24, 56, 25, 5]. Recent approaches [16, 34] have introduced cross-attention injection techniques to facilitate the editing of desired semantic regions on spatial feature maps. Subsequent works further improved upon this technique by incorporating semantic loss [28] or constraining the plugged feature with attention loss [55].

**Image Translation.**    Image translation aims to convert an image from one domain to another while preserving domain-irrelevant characteristics. Early studies [67, 30, 23, 60, 38] have employed conditional Generative Adversarial Networks (GANs) to ensure that the translated outputs conform to the distribution of the target domain. However, these approaches typically requires training a specific network for each translation task and relies on collections of instance images from both the source and target domains. Other approaches [1, 46, 53, 54, 57] have explored exploiting domain knowledge from pre-trained GANs or leveraging augmentation techniques with a single image to achieve image translation with limited data. Another slew of studies [70, 3, 59, 42, 22] focus on exemplar-based image translation due to their flexibility and superior image quality. While the majority of these approaches learn style transfer from a reference image, CoCosNet [72] proposes capturing fine structures from the exemplar image through dense correspondence learning.

**In-Context Learning.**    In-Context Learning [7], a concept originating from the realm of Natural Language Processing (NLP), offers an innovative approach to task completion. This paradigm achieves a given task by providing it with a set of sample examples alongside a query example, demonstrating exceptional proficiency in executing these tasks through a few-shot learning mechanism. Expanding on this notion, VisualPrompting [4] was a pioneering endeavor that introduced the concept of visual in-context learning. This framework demonstrated its remarkable effectiveness in various applications, including image segmentation, object detection, and colorization, all implemented within the framework of Masked Image Modeling (MIM). Painter [61] further refined and extended the MIM process, broadening its scope to encompass tasks like key point detection and image denoising. Their subsequent work SegGPT [62] delved into the exploration of various intricate tasks within the domain of image segmentation. Notably, there are very few works investigating visual in-context learning from the perspective of image generation. Thus, this paper explores the possibility of in-context image manipulation, attempting to promote novel applications functioning as ImageBrush.

## 3    Method

In this section, we will discuss the details of our proposed *Exemplar-Based Image Manipulation Framework*, **ImageBrush**. The primary objective of this framework is to develop a model capable of performing various image editing tasks by interpreting visual prompts as instructions. To achieve this, the model must possess the ability to comprehend the underlying human intent from contextual visual instructions and apply the desired operations to a new image. The entire pipeline is depicted in Fig. 2, where we employ a diffusion-based inpainting strategy to facilitate unified context learning and image synthesis. To further augment the model's reasoning capabilities, we meticulously design a visual prompt encoding module aimed at deciphering the human intent underlying the visual instruction.

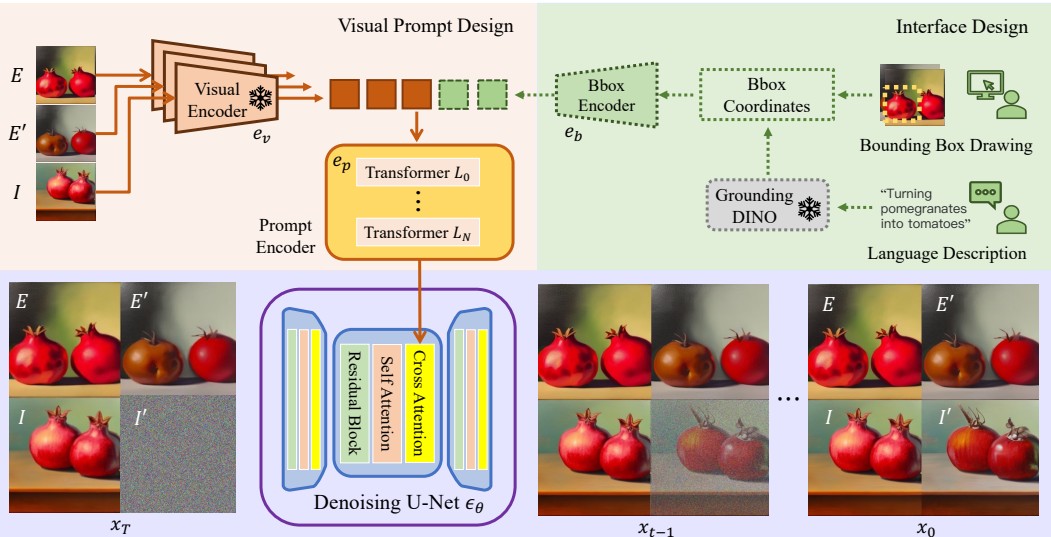

Figure 2: **Illustration of ImageBrush.** We introduce a novel and intuitive way of interacting with images. Users can easily manipulate images by providing *a pair of examples and a query image as prompts* to our system. If users wish to convey more precise instructions, they have the option to inform the model about their areas of interest through manual bounding box annotations or by using language to automatically generate them.

## 3.1 Exemplar-Based Image Manipulation

**Problem Formulation.** Given a pair of manipulated examples $\{\mathbf{E}, \mathbf{E}'\}$ and a query image $\mathbf{I}$, our training objective is to generate an edited image $\mathbf{I}'$ that adheres to the underlying instructions provided by the examples. Accomplishing this objective necessitates a comprehensive understanding of the intra-relationships between the examples as well as their inter-correlations with the new image. Unlike text, images are not inherently sequential and contain a multitude of semantic information dispersed across their spatial dimensions.

Therefore, we propose a solution in the form of "Progressive In-Painting". A common approach is to utilize cross-attention to incorporate the demonstration examples $\{\mathbf{E}, \mathbf{E}'\}$ and query image $\mathbf{I}$ as general context like previous work [66]. However, we observed that capturing detailed information among visual instructions with cross-attention injection is challenging. To overcome this, we propose learning their low-level context using a self-attention mechanism. Specifically, we concatenate a blank image $\mathbf{M}$ to the visual instructions and compose a grid-like image $\{\mathbf{E}, \mathbf{E}', \mathbf{I}, \mathbf{M}\}$ as illustrated in Fig. 2. The objective is to iteratively recover $\{\mathbf{E}, \mathbf{E}', \mathbf{I}, \mathbf{I}'\}$ from this grid-like image.

**Preliminary on Diffusion Model.** After an extensive examination of recent generative models, we have determined that the diffusion model [47] aligns with our requirements. This model stands out due to its stable training objective and exceptional ability to generate high-quality images. It operates by iteratively denoising Gaussian noise to produce the image $x_0$. Typically, the diffusion model assumes a Markov process [43] wherein Gaussian noises are gradually added to a clean image $x_0$ based on the following equation:

$$x_t = \sqrt{\alpha_t}x_0 + \sqrt{1 - \alpha_t}\epsilon, \tag{1}$$

where $\epsilon \sim \mathcal{N}(0, \mathbf{I})$ represents the additive Gaussian noise, $t$ denotes the time step and $\alpha_t$ is scalar functions of $t$. Our training objective is to devise a neural network $\epsilon_\theta(x_t, t, c)$ to predict the added noise $\epsilon$. Empirically, a simple mean-squared error is leveraged as the loss function:

$$L_{simple} := \mathbb{E}_{\epsilon \sim \mathcal{N}(0, \mathbf{I}), x_0, c}\left[\|\epsilon - \epsilon_\theta(x_t, c)\|_2^2\right], \tag{2}$$

where $\theta$ represents the learnable parameters of our diffusion model, and $c$ denotes the conditional input to the model, which can take the form of another image [49], a class label [20], or text [25]. By incorporating this condition, the classifier-free guidance [45] adjusts the original predicted noise $\epsilon_\theta(x_t, \varnothing)$ towards the guidance signal $\epsilon_\theta(x_t, c)$, formulated as

$$\hat{\epsilon}_\theta(x_t, c) = \epsilon_\theta(x_t, \varnothing) + w(\epsilon_\theta(x_t, c) - \epsilon_\theta(x_t, \varnothing)). \tag{3}$$

The $w$ is the guidance scale, determining the degree to which the denoised image aligns with the provided condition $c$.

**Context Learning by Progressive Denoising.** The overall generation process is visualized in Fig.2. Given the denoised result $x_{t-1} = \text{Grid}(\{\mathbf{E}, \mathbf{E}', \mathbf{I}, \mathbf{I}'\}_{t-1})$ from the previous time step $t$, our objective is to refine this grid-like image based on the contextual description provided in the visual instructions. Rather than directly operating in the pixel space, our model diffuses in the latent space of a pre-trained variational autoencoder, following a similar protocol to Latent Diffusion Models (LDM)[47]. This design choice reduces the computational resources required during inference and enhances the quality of image generation. Specifically, for an image $x_t$, the diffusion process removes the added Gaussian noise from its encoded latent input $z_t = \mathcal{E}(x_t)$. At the end of the diffusion process, the latent variable $z_0$ is decoded to obtain the final generated image $x_0 = \mathcal{D}(z_0)$. The encoder $\mathcal{E}$ and decoder $\mathcal{D}$ are adapted from Autoencoder-KL [47], and their weights are fixed in our implementation.

Unlike previous studies that rely on external semantic information [29, 6], here we focus on establishing spatial correspondence within image channels. We introduce a UNet-like network architecture, prominently composed of self-attention blocks, as illustrated in Fig. 2. This design enables our model to attentively process features within each channel and effectively capture their interdependencies.

## 3.2 Prompt Design for Visual In-Context Instruction Learning

However, relying only on universal correspondence modeling along the spatial channel may not be sufficient for comprehending abstract and complex concepts, which often require reassembling features from various aspects at multiple levels. To address this issue, we propose an additional prompt learning module to enable the model to capture high-level semantics without compromising the synthesis process of the major UNet architecture.

**Contextual Exploitation of Visual Prompt.** Given the visual prompt $vp = \{\mathbf{E}, \mathbf{E}', \mathbf{I}\}$, we aim to exploit their high-level semantic relationships. To achieve this, a visual prompt module comprising two components is carefully devised, which entails a shared visual encoder $e_v$ and a prompt encoder $e_p$ as illustrated in Fig. 2. For an arbitrary image $\mathbf{I} \in \mathbb{R}^{H \times W \times 3}$ within the visual prompt, we extract its tokenized feature representation $f_{img}$ using the Visual Transformer (ViT)-based backbone $e_v$. These tokenized features are then fed into the bi-directional transformer $e_p$, which effectively exploits the contextual information. The resulting feature representation $f_c$ encapsulates the high-level semantic changes and correlations among the examples.

Using the visual prompt, we can integrate high-level semantic information into the UNet architecture by employing the classifier-free guidance strategy as discussed in Section 3.1. This is achieved by injecting the contextual feature into specific layers of the main network through cross-attention.

$$\phi^{l-1} = \phi^{l-1} + \text{Conv}(\phi^{l-1}) \tag{4}$$

$$\phi^{l-1} = \phi^{l-1} + \text{SelfAttn}(\phi^{l-1}) \tag{5}$$

$$\phi^{l} = \phi^{l-1} + \text{CrossAttn}(f_c) \tag{6}$$

where $\phi^l$ denotes the input feature of the $l$-th block, and $f_c$ represents the context feature. Specifically, we select the middle blocks of the UNet architecture, as these blocks have been shown in recent studies to be responsible for processing high-level semantic information [58].

**Interface Design for Enhancing User Intent Comprehension.** Recent advancements in visual instruction systems have have showcased remarkable capabilities through the integration of human prompts [27]. Taking inspiration from this research, we introduce a similar interface module that empowers users to emphasize their areas of interest. This, in turn, enhances the system's understanding of user intention to some extent.

Users have the option to designate their area of interest either by manually drawing a bounding box or by utilizing automated tools [31], as depicted in Fig. 2. Similar to [29], these selected boxes are first processed by a box encoder, denoted as $e_b$, before being integrated into the system. During the training phase, we employ GroundingDINO [31] to label the focused region based on textual instruction. This approach not only alleviates the burden of manual labeling but also offers users the flexibility to opt for an automatic tool that leverages language descriptions to enhance their intentions, particularly in cases where drawing a bounding box is less preferable. Additionally, we randomly remove a portion of the bounding boxes to ensure that our model fully harnesses the benefits of the visual in-context instructions.

Table 1: **Quantitative Comparison on In-the-wild Dataset.**

| Dataset | Exemplar-Based Image Translation | | | Pose Transfer | | Inpainting | | | Editing |
|---|---|---|---|---|---|---|---|---|---|
| | Scannet[12] | LRW(Edge)[10] | LRW(Mask)[10] | UBC-Fashion[69] | | DAVIS[41] | | | InstructPix2Pix-Filtered[6] |
| Metric | FID↓ | FID↓ | FID↓ | FID↓ | SSIM↑ | FID↓ | V-FID↓ | SSIM↑ | SSIM↑ |
| TSAM[73] | - | - | - | - | - | 86.84 | 158.30 | **0.901** | - |
| CoCosNet[72] | 19.49 | 15.44 | 14.25 | 38.61 | 0.885 | - | - | - | - |
| VisualPrompt[4] | - | - | - | - | 0.677 | - | - | 0.787 | 0.417 |
| **ImageBrush** | **9.18** | **9.67** | **8.95** | **12.99** | **0.910** | **18.70** | 175.25 | 0.816 | **0.437** |

## 4 Experiments

### 4.1 Experimental Settings

**Datasets.** Our work leverages four widely used in-the-wild video datasets - Scannet [12], LRW [10], UBCFashion [69], and DAVIS [41] - as well as a synthetic dataset that involves numerous image editing operations. The **Scannet** [12] dataset is a large-scale collection of indoor scenes covering various indoor environments, such as apartments, offices, and hotels. It comprises over 1,500 scenes with rich annotations, of which 1,201 scenes lie in training split, 312 scenes are in the validation set. No overlapping physical location has been seen during training. The **LRW** [10] dataset is designed for lip reading, containing over 1000 utterances of 500 different words with a duration of 1-second video. We adopt 80 percent of their test videos for training and 20 percent for evaluation. The **UBC-Fashion** [69] dataset comprises 600 videos covering a diverse range of clothing categories. This dataset includes 500 videos in the training set and 100 videos in the testing set. No same individuals has been seen during training. The **DAVIS** [41] (Densely Annotated VIdeo Segmentation) dataset is a popular benchmark dataset for video object segmentation tasks. It comprises a total of 150 videos, of which 90 are densely annotated for training and 60 for validation. Similar to previous works [73], we trained our network on the 60 videos in the validation set and evaluate its performance on the original 90 training videos. The synthetic image manipulation dataset is created using image captions of LAION Improved Aesthetics 6.5+ through Stable Diffusion. We use the CLIP-filtered subset processed by InstructPix2Pix [6]. It comprises over 310k editing instructions, each with its corresponding editing pairs. Out of these editing instructions, 260k have more than two editing pairs. We conduct experiment on this set, reserving 10k operations for model validation.

**Implementation Details.** In our approach, all input images have a size of $256 \times 256$ pixels and are concatenated as input to the UNet. The UNet architecture, adapted from Stable Diffusion, consists of 32 blocks with self-attention and cross-attention layers. We use cross-attention to incorporate the features of the visual prompt module into its two middle blocks. The visual prompt's shared backbone, denoted as $e_v$, follows the architecture of EVA-02 [13]. The prompt encoder $e_p$ comprises five layers and has a latent dimension of 768. For the bounding box encoder, denoted as $e_b$, we adopt a simple MLP following [29]. During training, we set the classifier-free scale for the encoded instruction to 7.5 and the dropout ratio to 0.05. Our implementation utilizes PyTorch [39] and is trained on 24 Tesla V100-32G GPUs for 14K iterations using the AdamW [26] optimizer. The learning rate is set to $1e$-6, and the batch size is set to 288.

**Comparison Methods.** To thoroughly evaluate the effectiveness of our approach, we compare it against other state-of-art models tailored for specific tasks, including: **VisualPrompt** [4], the pioneer study in visual in-context learning; **SDEdit** [33], a widely-used stochastic differential equation (SDE)-based image editing method; **Instruct-Pix2pix** [6], a cutting-edge language-instructed image manipulation method; **CoCosNet** [72], an approach that employs a carefully designed correspondence learning architecture to achieve exemplar-based image translation; and **TSAM** [73], a model that incorporates a feature alignment module for video inpainting.

### 4.2 Quantitative Evaluation

**Evaluation Metrics.** In the image manipulation task, it is crucial for the edited image to align with the intended direction while preserving the instruction-invariant elements in their original form. To assess the degree of agreement between the edited image and the provided instructions, we utilize a cosine similarity metric referred to as **CLIP Direction Similarity** in the CLIP space. In order to measure the level of consistency with original image, we utilize the cosine similarity of the CLIP image embedding, **CLIP Image Similarity**, following the protocol presented in [6]. Additionally, for other generation tasks such as exemplar-based image translation, pose transfer, and video in-painting, we employ the Fréchet Inception Score (**FID**) [18] as an evaluation metric. The FID score allows

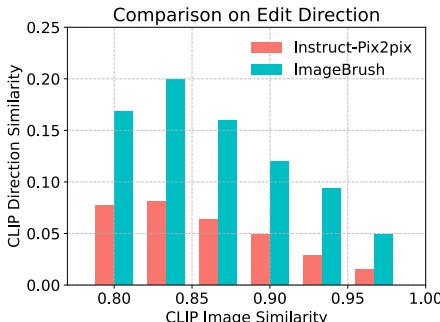
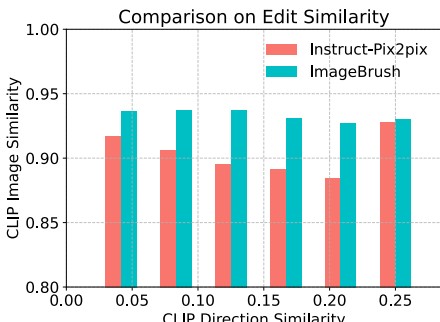

Figure 3: **Quantitative Comparison on Image Editing.** We compare our approach with the representative text-instruction method in terms of direction consistency and similarity consistency.

us to evaluate the dissimilarity between the distributions of synthesized images and real images. Following similar protocol with [73], the **V-FID** is also included to take into account the temporal consistency in video inpainting. For image generation, we introduce the **SSIM** [63] metric to further assess the structural similarity and visual quality.

**Image Generation Paradigm.** In the image manipulation task, a pair of transformed images is utilized as visual instructions to edit a target image within the same instruction context. For other tasks, we conduct experiments in a cross-frame prediction setting, where we select two temporally close frames and use one of them as an example to predict the other. Specifically, we employ one frame along with its corresponding label (e.g., edge, semantic segmentation, keypoints, or masked image) as a contextual example, and take the label from another frame as a query to recover that frame. To obtain the label information, we extract keypoints from the UBC-Fashion dataset using OpenPose [8], and for the LRW dataset, we utilize a face parsing network [68] to generate segmentation labels. To ensure that the selected frames belong to the same context, we restrict the optical flow between them to a specific range, maintaining consistency for both training and testing. It is important to note that, for fair comparison, we always utilize the first frame of each video as a reference during the evaluation of the video inpainting task.

**Evaluation Results.** The comparison results for image manipulation are presented in Fig. 3. We observe that our results exhibit higher image consistency for the same directional similarity values and higher directional similarity values for the same image similarity value. One possible reason is the ability of visual instructions to express certain concepts without a modality gap, which allows our method to better align the edited images.

To demonstrate the versatility of our model, we conducted experiments using in-the-wild videos that encompassed diverse real-world contexts. The results for various downstream tasks are presented in Table 1, showing the superior performance of our approach across most datasets. It is noteworthy that our model achieves these results using a single model, distinguishing it from other methods that require task-specific networks. In inpainting task, our model attains a higher FID score while yielding a comparatively lower Video-FID score and SSIM value. We speculate that it is because our framework lack explicit temporal awareness such as optical flow in [73].

## 4.3 Qualitative Evaluation

We provide a qualitative comparison with SDEdit [33] and Instruct-Pix2pix [6] in Fig. 4. In the case of the SDEdit model, we attempted to input both the edit instruction and the output caption, referred to as SDEdit-E and SDEdit-OC, respectively. In contrast to language-guided approaches, our method demonstrates superior fidelity to the provided examples, particularly in capturing text-ambiguous concepts such as holistic style, editing extent, and intricate local object details. Comparison with VisualPrompt [4] is depicted in Fig. 6 where we achieve more realistic and plausible editing results.

Additionally, we provide a qualitative analysis on a real-world dataset in Fig. 5. Compared to CocosNet [72], our approach demonstrates superior visual quality in preserving object structures, as seen in the cabinet and table. It also exhibits higher consistency with the reference appearance, particularly noticeable in the woman's hair and dress. Furthermore, our method achieves improved shape consistency with the query image, as observed in the mouth. When compared to TSAM [73], our

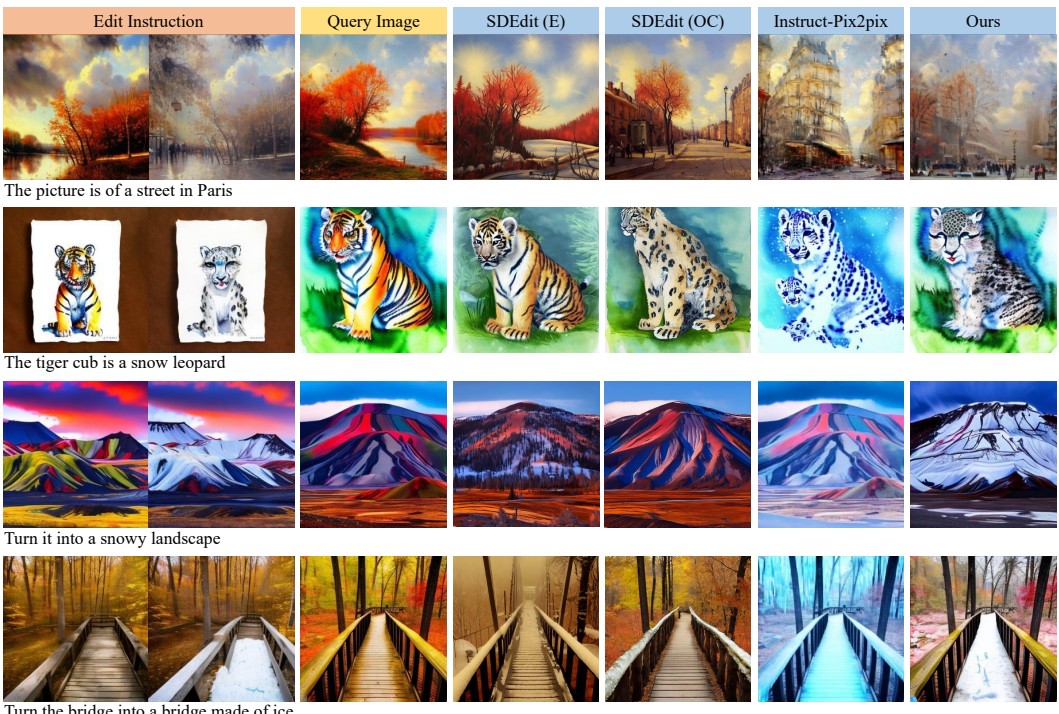

Figure 4: **Qualitative Comparison on Image Manipulation**. In contrast to language-guided approaches, our editing results guided by visual instruction exhibit better compliance with the provided examples.

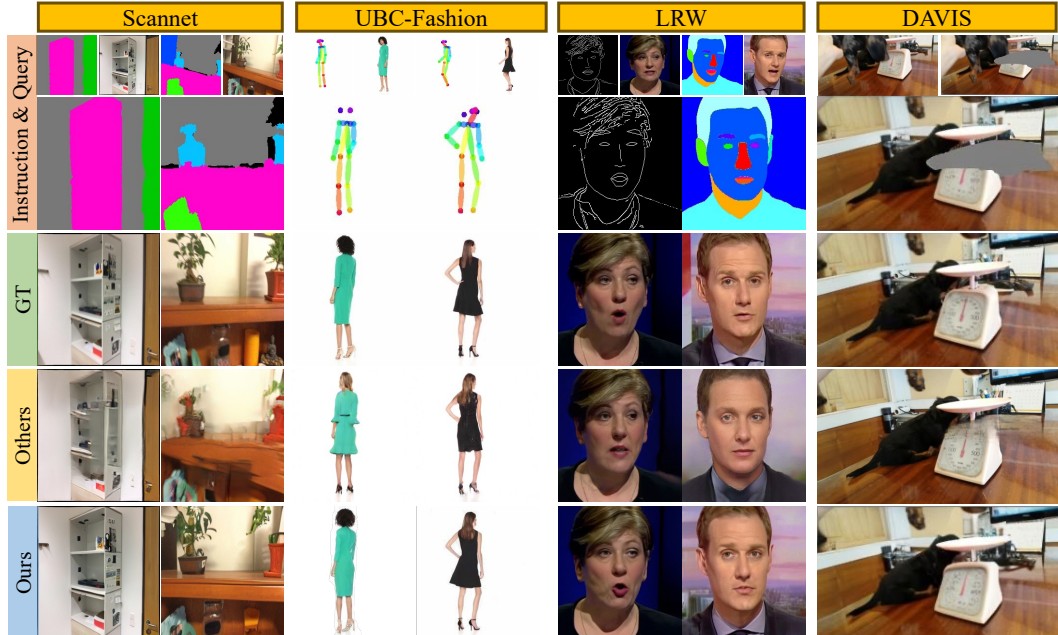

Figure 5: **Qualitative Comparison on In-the-wild Dataset.** We conduct experiments on various downstream tasks including exemplar-based image translation, pose transfer and inpainting.

approach yields superior results with enhanced clarity in object shapes and more realistic background filling. These improvements are achieved by effectively leveraging visual cues from the examples.

### 4.4 Further Analysis

**Novel Evaluation Metric.** Exemplar-based image manipulation is a novel task, and the evaluation metric in the literature is not suitable for this task. Therefore, we present an evaluation metric

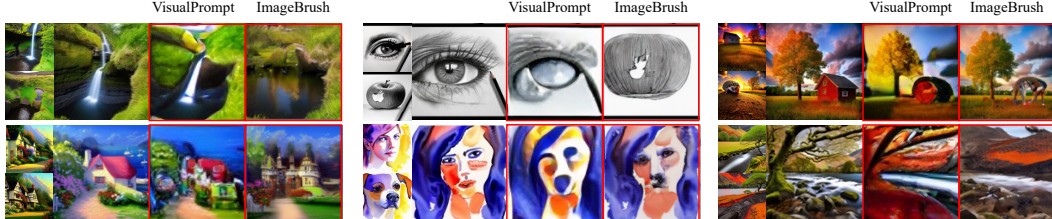

Figure 6: **Comparison with VisualPrompt [4] on Image Editing.** Our model achieves plausible and realistic edits whereas VisualPrompt tend to produce conservative prediction and struggle on instruction comprehension.

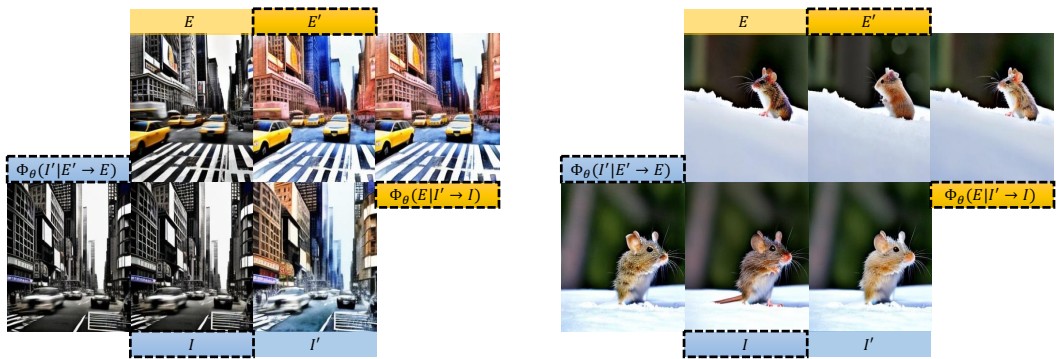

Figure 7: **Case Study in Terms of Prompt Fidelity and Image Fidelity.** The prompt fidelity and image fidelity are calculated between paired images with dashed yellow banner and dashed blue banner, respectively.

that requires neither ground-truth image nor human evaluation. Specifically, our model, denoted as $\Phi_\theta$, takes the examples $\{\mathbf{E}, \mathbf{E}'\}$, and the source image $\mathbf{I}$ to produce a manipulated output $\mathbf{I}'$. This procedure is presented as $\mathbf{I}' = \Phi_\theta(\mathbf{I}|\mathbf{E}' \to \mathbf{E})$. The goal is to let the output image $\mathbf{I}'$ abide by instruction induced from examples $\mathbf{E}$ and $\mathbf{E}'$ while preserving the instruction-invariant content of input image $\mathbf{I}$. We define prompt fidelity to assess the model's ability to follow the instruction. According to the symmetry of $\{\mathbf{E}, \mathbf{E}'\}$ and $\{\mathbf{I}, \mathbf{I}'\}$, if its operation strictly follows the instruction, the model should manipulate $\mathbf{E}$ to $\mathbf{E}'$ by using $\mathbf{I} \to \mathbf{I}'$ as the prompt (see pictures labeled with a dashed yellow tag in Fig. 7). We express the prompt fidelity as follows

$$\Delta_{\text{prompt}} = \text{FID}(\mathbf{E}', \Phi_\theta(\mathbf{E}|\mathbf{I} \to \mathbf{I}')). \tag{7}$$

On the other hand, to evaluate the extent to which the model preserves its content, we introduce an image fidelity metric. If $\mathbf{I}'$ maintains the content of $\mathbf{I}$, the manipulation should be invertible. That is to say, by using $\mathbf{E}' \to \mathbf{E}$ as the prompt, the model should reconstruct $\mathbf{I}$ from $\mathbf{I}'$ (see pictures labeled with a dashed blue tag in Fig. 7). Therefore, we define the image fidelity as

$$\Delta_{\text{image}} = \text{FID}(\mathbf{I}, \Phi_\theta(\mathbf{I}'|\mathbf{E}' \to \mathbf{E})). \tag{8}$$

**Ablation Study.** We performed ablation studies on three crucial components of our method, namely the diffusion process for context exploitation, the vision prompt design, and the injection of human-interest area. Specifically, we conducted experiments on our model by (1) replacing the diffusion process with masked image modeling, (2) removing the cross-attention feature injection obtained from vision prompt module, and (3) deactivating the input of the human interest region. Specifically, we implemented the masked image modeling using the MAE-VQGAN architecture following the best setting of [4]. The numerical results on image manipulation task are shown in Table 2. The results demonstrate the importance of the diffusion process. Without it, the model is unable to progressively comprehend the visual instruction and refine its synthesis, resulting in inferior generation results. When we exclude the feature integration from the visual prompt module, the model struggles to understand high-level semantics, leading to trivial generation results on these instructions. Furthermore, removing the region of interest interface results in a slight decrease in performance, highlighting its effectiveness in capturing human intent.

**Case Study.** In Fig. 7, we present two cases that showcase our model's performance in terms of image and prompt fidelity. The first case, depicted on the left, highlights our model's capability to

Table 2: **Ablation Study on Image Manipulation.** Here we evaluate on our proposed metric.

|  | w/o Diffusion | w/o Cross-Attention | w/o Interest Region | Full Model |
|---|---|---|---|---|
| Prompt Fidelity | 78.65 | 39.44 | 24.29 | **23.97** |
| Image Fidelity | 82.33 | 41.51 | 25.74 | **24.43** |

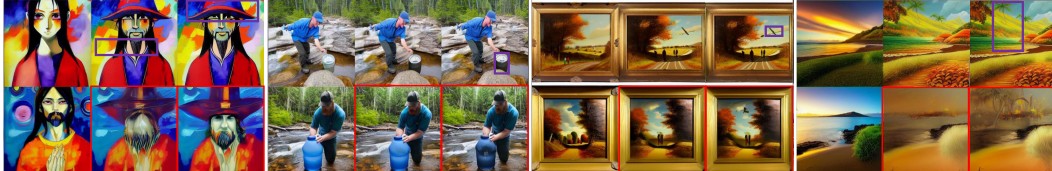

Figure 8: **Generation Results by Customized Prompts.** The regions of interest are indicated by purple box and generated results are displayed within red box. From left to right we illustrate cases where emphasis is placed on beards, hats, buckets, flying objects and trees.

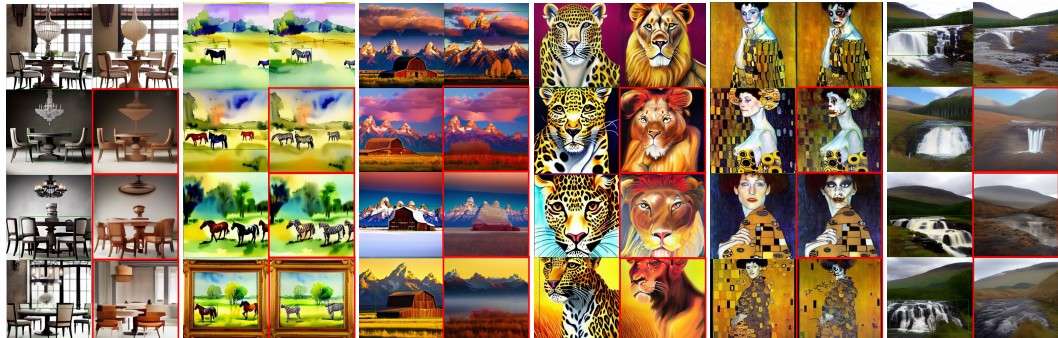

Figure 9: **Image Editing with Varied Queries.** We showcase edited results corresponding to three distinct queries. It covers a variety of query contexts, demonstrating changes in layout, object quantity, weather, pose, shape and perspective.

utilize its predicted result $\mathbf{I}'$ to reconstruct $\mathbf{I}$ by transforming the image back to a dark appearance. With predicted $\mathbf{I}'$, our model also successfully edits the corresponding example from $\mathbf{E}$ to $\mathbf{E}'$, making the example image colorful. On the right side of the figure, we present a case where our model encounters difficulty in capturing the change in the background, specifically the disappearance of the holographic element.

**Effect of User Interface.** Fig. 8 depicts diverse examples with distinct emphasis on visual instructions. For example, synthesis of additional beards when the beard area is emphasized, the generation of more detailed hats when a bounding box is applied to the hat part, and the improved ability of our model to understand and execute the intention of editing a bucket to black when the emphasis is on the bucket. Furthermore, when a small object is added, our model successfully comprehends and synthesizes an object with the assistance of this input. Similarly, by placing emphasis on the background, more trees are synthesized in that region.

**Image Editing with Diverse Queries.** We present a collection of qualitative examples encompassing diverse query contexts in Fig. 9. The results suggest our model's competence in generating coherent and reliable outcomes across various query scenarios.

## 5   Conclusion

In this article, we present an innovative way of interacting with images, *Image Manipulation by Visual Instruction*. Within this paradigm, we propose a framework, **ImageBrush**, which holds promising potential for various applications.

**Limitation and Future Work.** 1) The proposed model may face challenges when there is a substantial disparity between the given instruction and the query image. 2) The current model also encounters difficulties in handling intricate details, such as subtle background changes or adding small objects. 3) Future work can delve into a broader range of datasets and tasks, paving the way for future generative visual foundation models.

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

# Appendices

## A    Analysis of In-Context Instruction with Multiple Examples

Our approach can be naturally extended to include multiple examples. Specifically, given a series of examples $\{\mathbf{E_1}, \mathbf{E'_1}, \ldots, \mathbf{E_n}, \mathbf{E'_n}, \mathbf{I}\}$, where $n$ represents the number of support examples, our objective is to generate $\mathbf{I}'$. In our main paper, we primarily focused on the special case where $n = 1$. When dealing with multiple examples, we could also establish their spatial correspondence by directly concatenating them as input to our UNet architecture. Specifically, we create a grid $x_{t-1} = \mathrm{Grid}(\{\mathbf{E_1}, \mathbf{E'_1}, \ldots, \mathbf{E_n}, \mathbf{E'_n}, \mathbf{I}\}_{t-1})$ that can accommodate up to eight examples following [4]. To facilitate contextual learning, we extend the input length of our prompt encoder $e_p$, and incorporate tokenized representations of these collections of examples as input to it. In this way, our framework is able to handle cases that involve multiple examples.

Below we discuss the impact of these examples on our model's final performance by varying their numbers and orders.

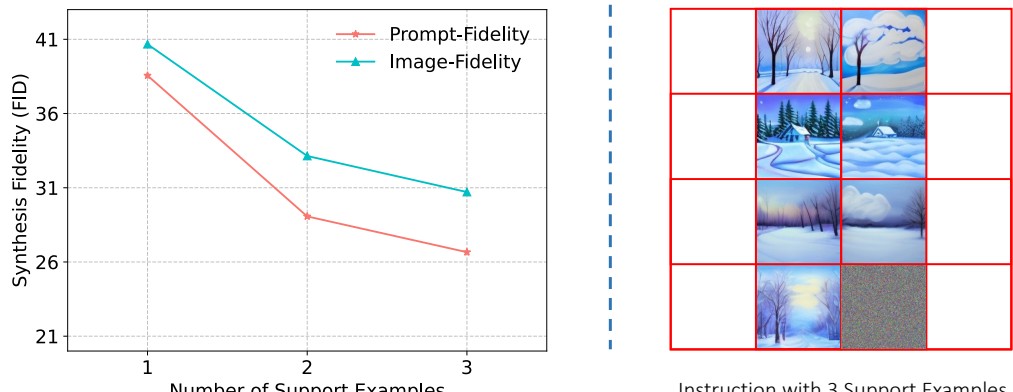

Figure 10: **Analysis of In-Context Instruction with Multiple Examples.**

### A.1    Number of In-Context Examples.

In our dataset, which typically consists of 4 examples, we examine the influence of the number of in-context examples by varying it from 1 to 3, as illustrated in Figure 10. We evaluate this variation using our proposed metrics: prompt fidelity $\Delta_{\mathrm{prompt}}$ and image fidelity $\Delta_{\mathrm{image}}$, which assess instruction adherence and content consistency, respectively. The results demonstrate that increasing the number of support examples enhances performance, potentially by reducing task ambiguity.

Furthermore, we present additional instruction results in Figure 11, showcasing the impact of different numbers of in-context examples. It is clear that as the number of examples increases, the model becomes more confident in comprehending the task description. This is particularly evident in the second row, where the synthesized floor becomes smoother and the rainbow appears more distinct. Similarly, in the third row, the wormhole becomes complete. However, for simpler tasks that can be adequately induced with just one example (as demonstrated in the last row), increasing the number of examples does not lead to further performance improvements.

### A.2    Order of In-Context Examples.

We have also conducted an investigation into the impact of the order of support examples by shuffling them. Through empirical analysis, we found no evidence suggesting that the order of the examples has any impact on performance. This finding aligns with our initial expectation, as there is no sequential logic present in our examples.

## B    More Visual Instruction Results

In this section, we present additional visual instruction results, where the synthesized result is highlighted within the red box. These results demonstrate the effectiveness of our approach in handling diverse manipulation types, including style transfer, object swapping, and composite operations, as showcased in Fig. 12 and Fig. 13. Furthermore, our method demonstrates its versatility across real-world datasets, successfully tackling various downstream tasks, such as image translation, pose translation, and inpainting in Fig. 14 and Fig. 15. Importantly, it is worth noting that all the presented results are generated by a single model.

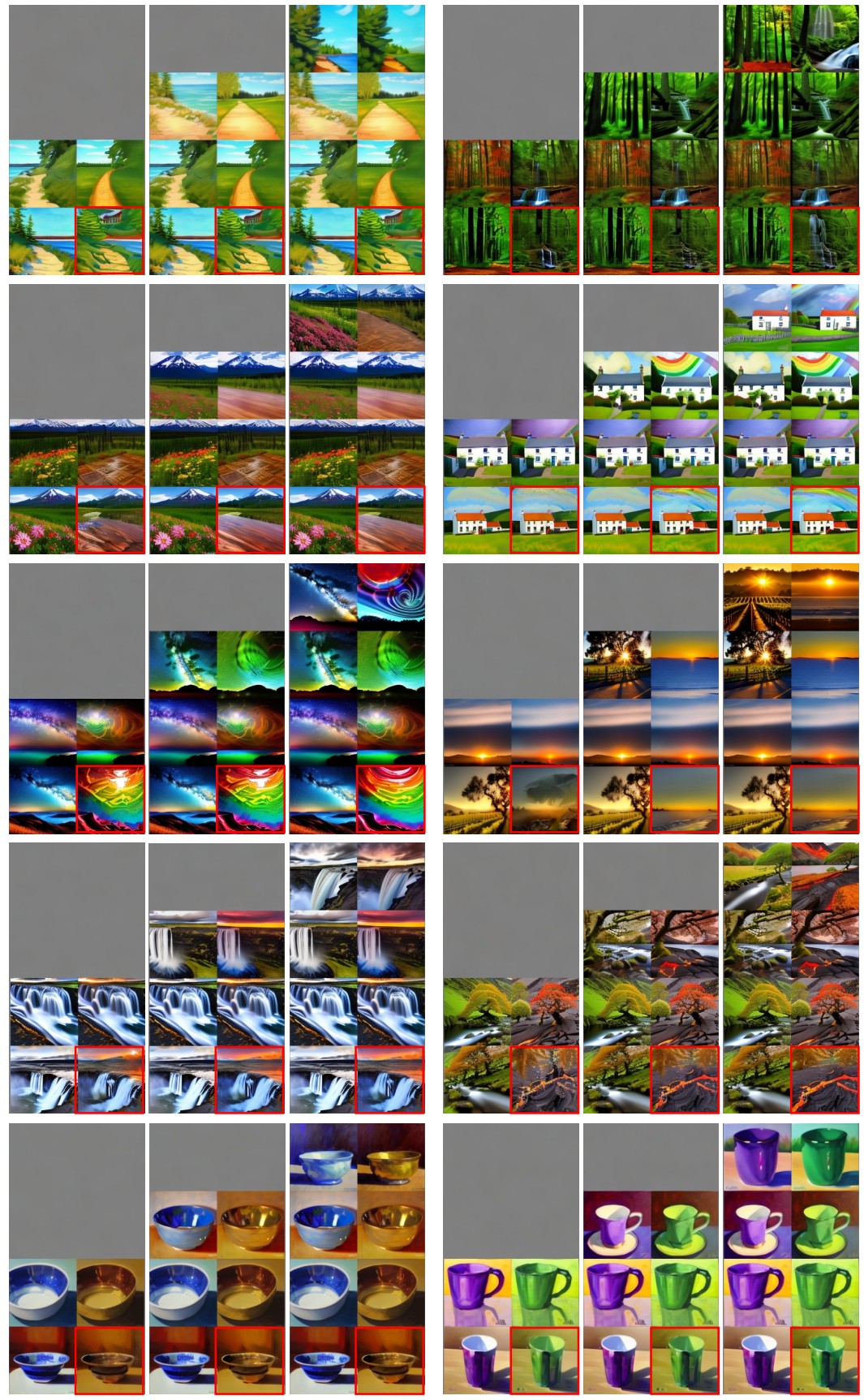

Figure 11: **Qualitative Analysis on Number of In-Context Examples.**

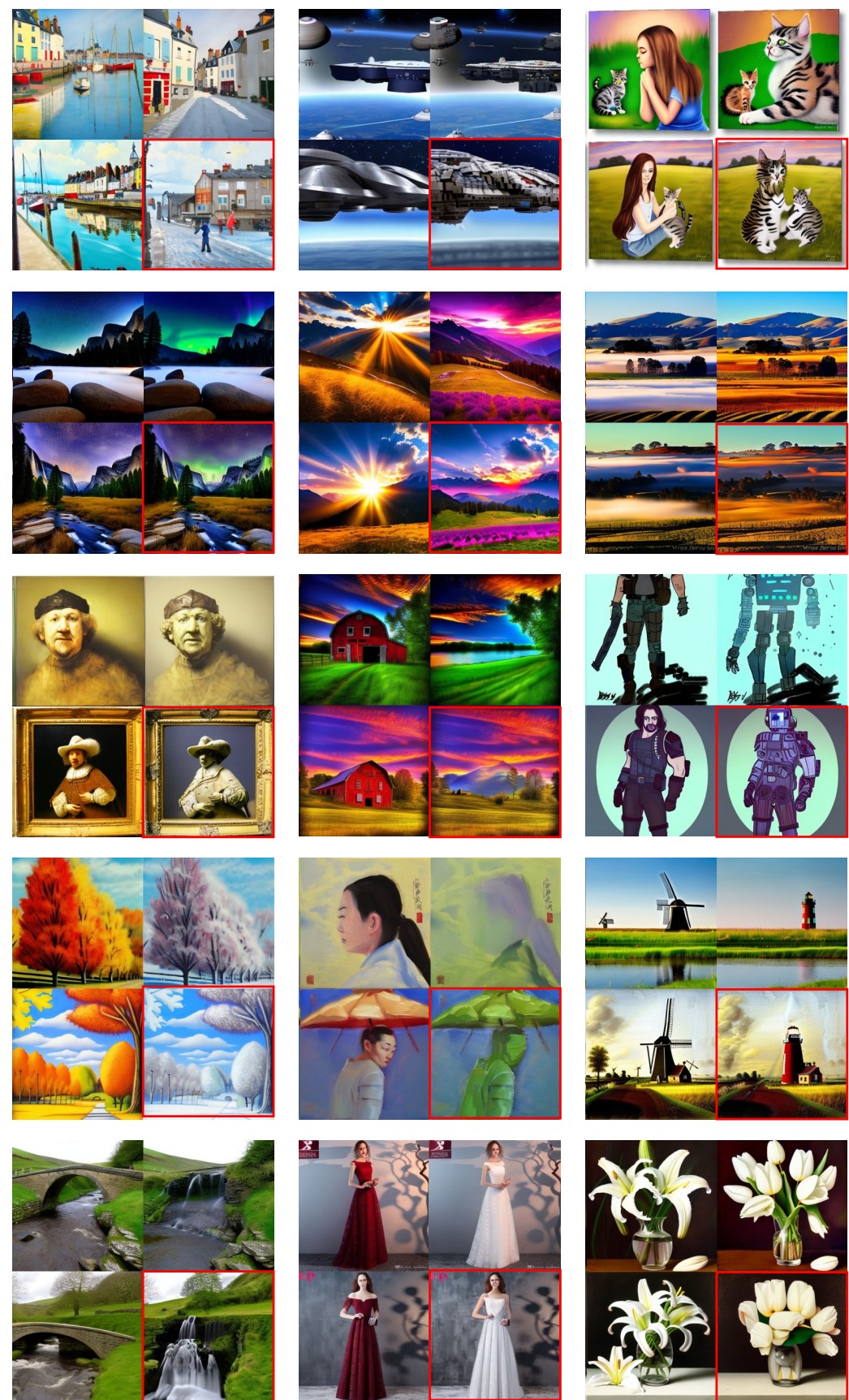

Figure 12: **Image Manipulation Results by Visual Instruction.**

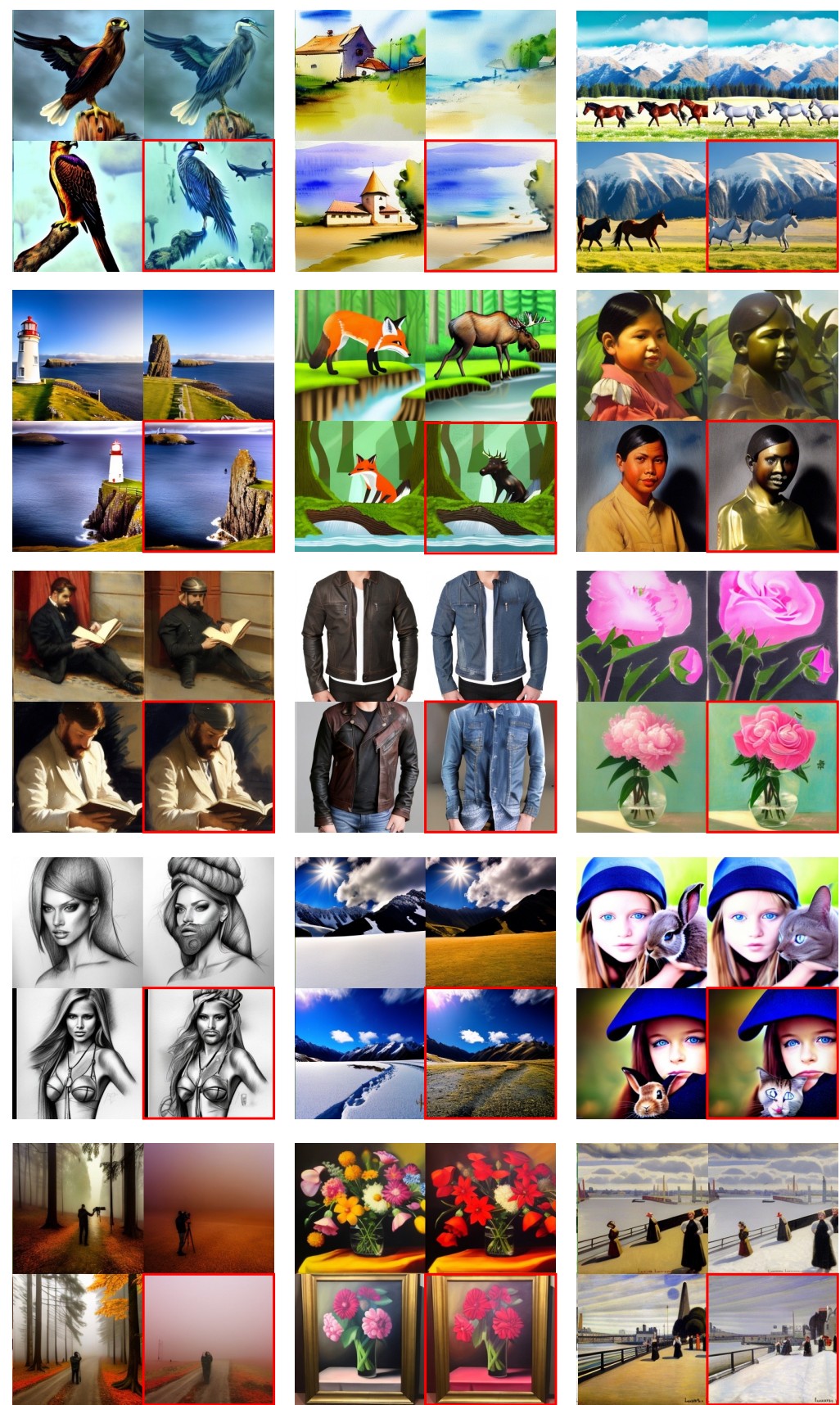

Figure 13: **Image Manipulation Results by Visual Instruction.**

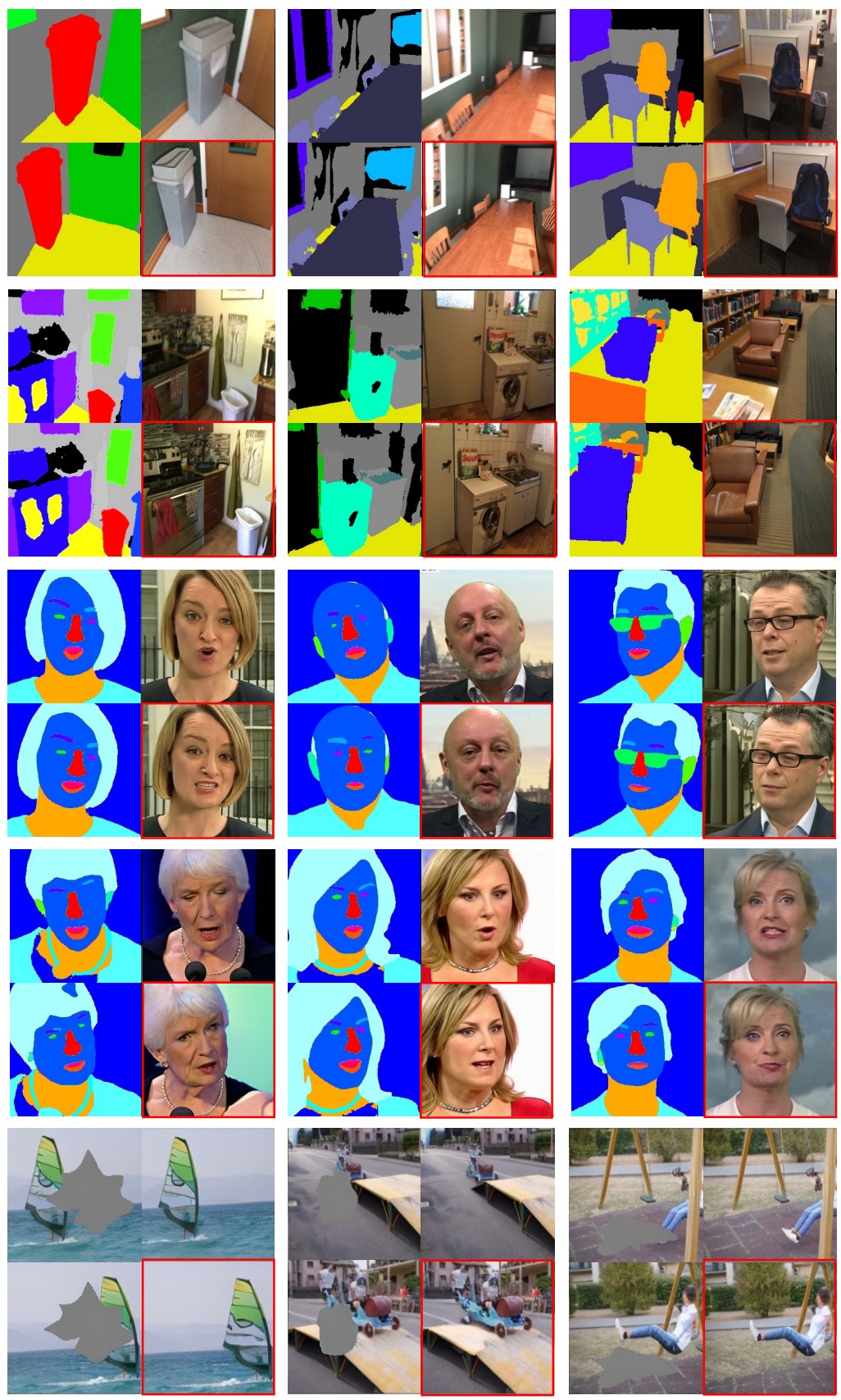

Figure 14: **Instruction Results on In-the-wild Dataset.**

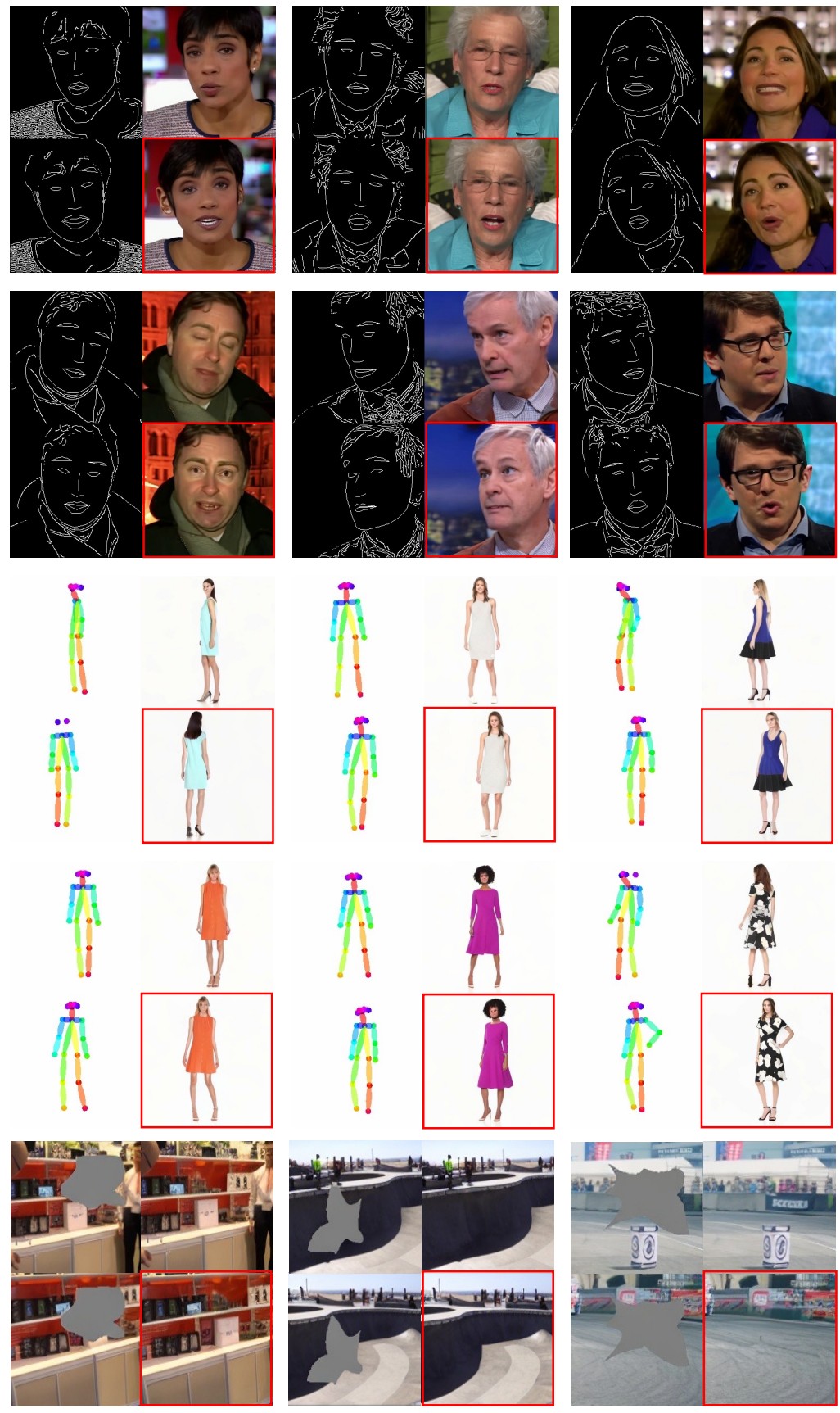

Figure 15: **Instruction Results on In-the-wild Dataset.**

