# ImageBrush: Learning Visual In-Context Instructions for Exemplar-Based Image Manipulation —— Supplementary Materials ——

## 1 Analysis of In-Context Instruction with Multiple Examples

Our approach can be naturally extended to include multiple examples. Specifically, given a series of examples $\{\mathbf{E_1}, \mathbf{E'_1}, \ldots, \mathbf{E_n}, \mathbf{E'_n}, \mathbf{I}\}$, where $n$ represents the number of support examples, our objective is to generate $\mathbf{I'}$. In our main paper, we primarily focused on the special case where $n = 1$. When dealing with multiple examples, we could also establish their spatial correspondence by directly concatenating them as input to our UNet architecture. Specifically, we create a grid $x_{t-1} = \text{Grid}(\{\mathbf{E_1}, \mathbf{E'_1}, \ldots, \mathbf{E_n}, \mathbf{E'_n}, \mathbf{I}\}_{t-1})$ that can accommodate up to eight examples following [1]. To facilitate contextual learning, we extend the input length of our prompt encoder $e_p$, and incorporate tokenized representations of these collections of examples as input to it. In this way, our framework is able to handle cases that involve multiple examples.

Below we discuss the impact of these examples on our model's final performance by varying their numbers and orders.

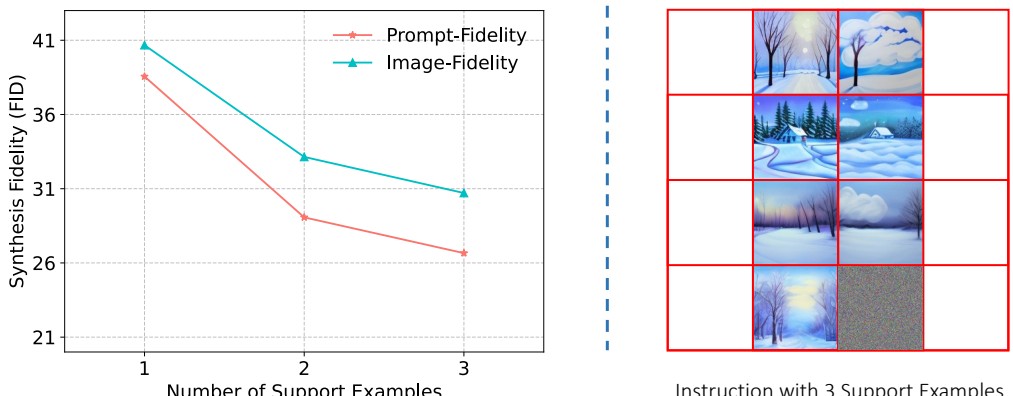

Figure 1: **Analysis of In-Context Instruction with Multiple Examples.**

### 1.1 Number of In-Context Examples.

In our dataset, which typically consists of 4 examples, we examine the influence of the number of in-context examples by varying it from 1 to 3, as illustrated in Figure 1. We evaluate this variation using our proposed metrics: prompt fidelity $\Delta_{\text{prompt}}$ and image fidelity $\Delta_{\text{image}}$, which assess instruction adherence and content consistency, respectively. The results demonstrate that increasing the number of support examples enhances performance, potentially by reducing task ambiguity.

Furthermore, we present additional instruction results in Figure 3, showcasing the impact of different numbers of in-context examples. It is clear that as the number of examples increases, the model

Submitted to 37th Conference on Neural Information Processing Systems (NeurIPS 2023). Do not distribute.

becomes more confident in comprehending the task description. This is particularly evident in the second row, where the synthesized floor becomes smoother and the rainbow appears more distinct. Similarly, in the third row, the wormhole becomes complete. However, for simpler tasks that can be adequately induced with just one example (as demonstrated in the last row), increasing the number of examples does not lead to further performance improvements.

## 1.2 Order of In-Context Examples.

We have also conducted an investigation into the impact of the order of support examples by shuffling them. Through empirical analysis, we found no evidence suggesting that the order of the examples has any impact on performance. This finding aligns with our initial expectation, as there is no sequential logic present in our examples.

## 2 Case Study on Visual Prompt User Interface

In our work, we have developed a human interface to further enhance our model's ability to understand human intent. Since manual labeling of human-intended bounding boxes is not available, we utilize language-grounded boxes [2] to train the network in our experiments. In Figure 2, we demonstrate the difference when the bounding box is utilized during inference. We can observe that when the crown area is further emphasized by bounding box, the synthesized crown exhibits a more consistent style with the provided example. Additionally, the dress before the chest area is better preserved.

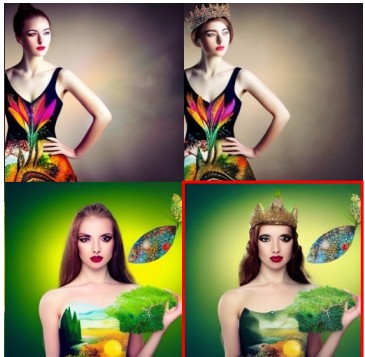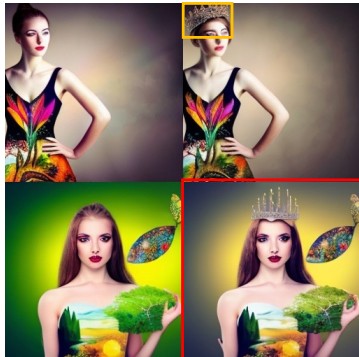

Figure 2: **Analysis of Visual Prompt User Interface.** On the left side, we illustrate the usual case, while on the right side, we present a scenario where a bounding box (indicated by the yellow box) is grounded by DINO as the model input.

## 3 More Visual Instruction Results

In this section, we present additional visual instruction results, where the synthesized result is highlighted within the red box. These results demonstrate the effectiveness of our approach in handling diverse manipulation types, including style transfer, object swapping, and composite operations, as showcased in Fig. 4 and Fig. 5. Furthermore, our method demonstrates its versatility across real-world datasets, successfully tackling various downstream tasks, such as image translation, pose translation, and inpainting in Fig. 6 and Fig. 7. Importantly, it is worth noting that all the presented results are generated by a single model.

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

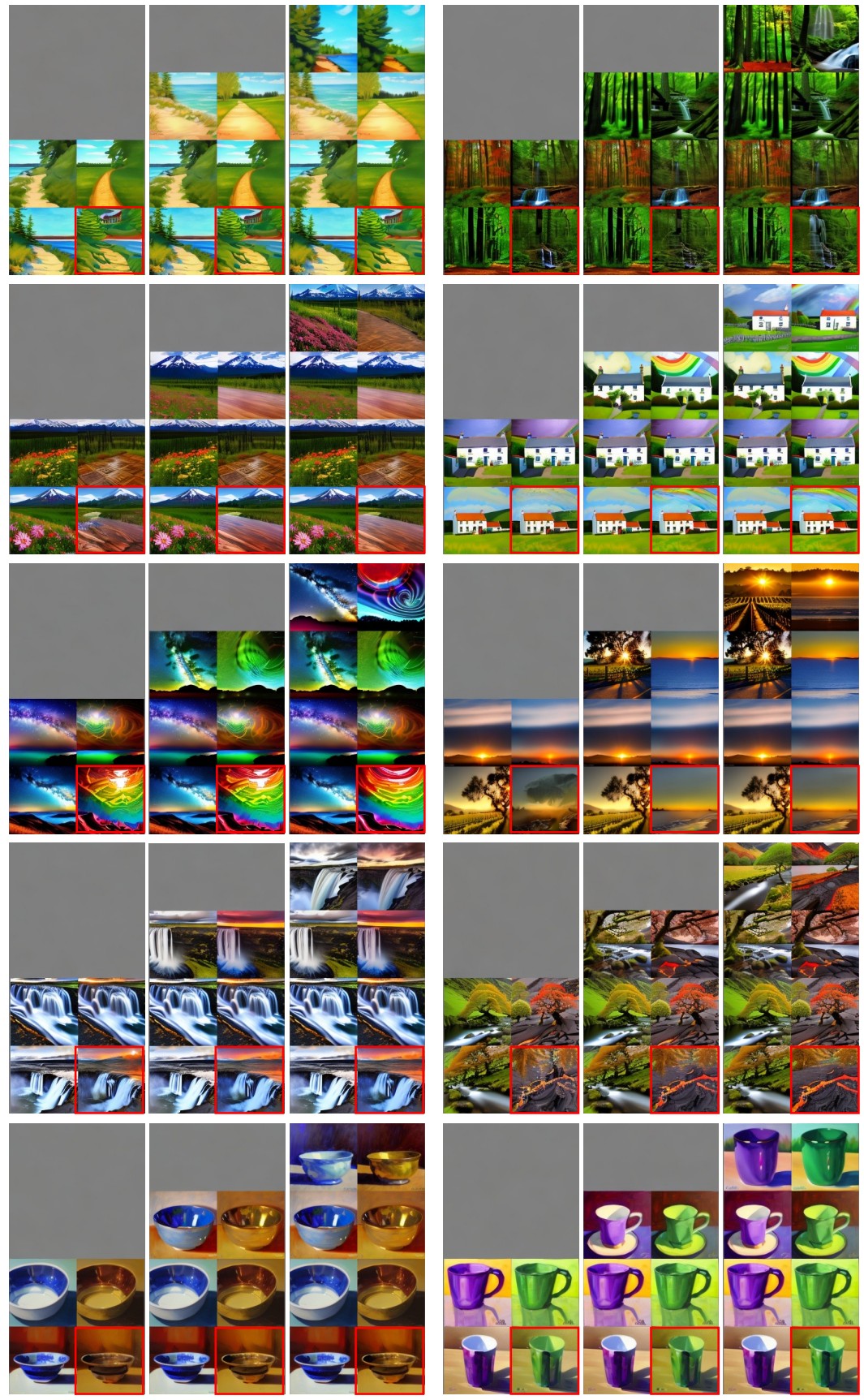

Figure 3: **Qualitative Analysis on Number of In-Context Examples.**

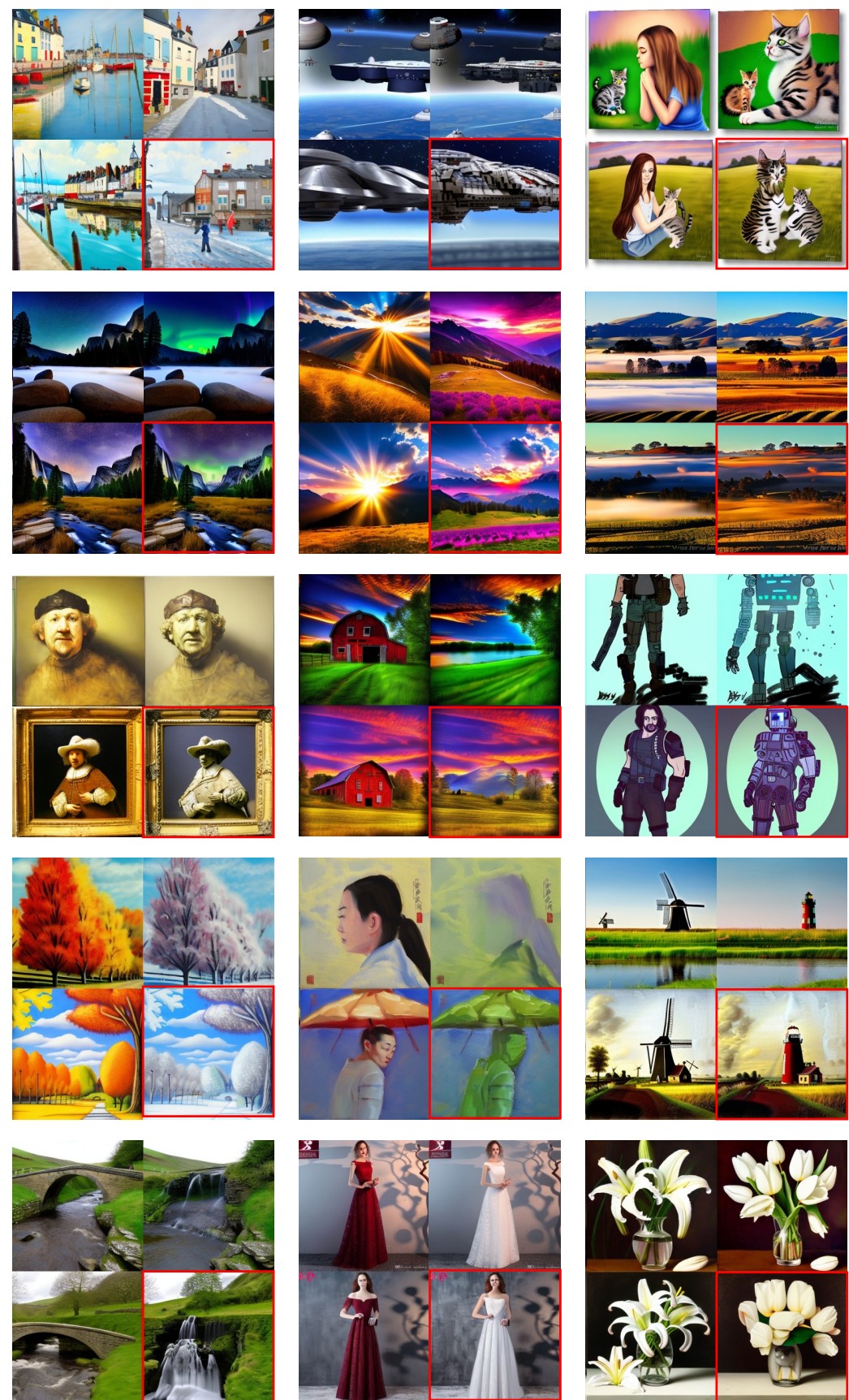

Figure 4: **Image Manipulation Results by Visual Instruction.**

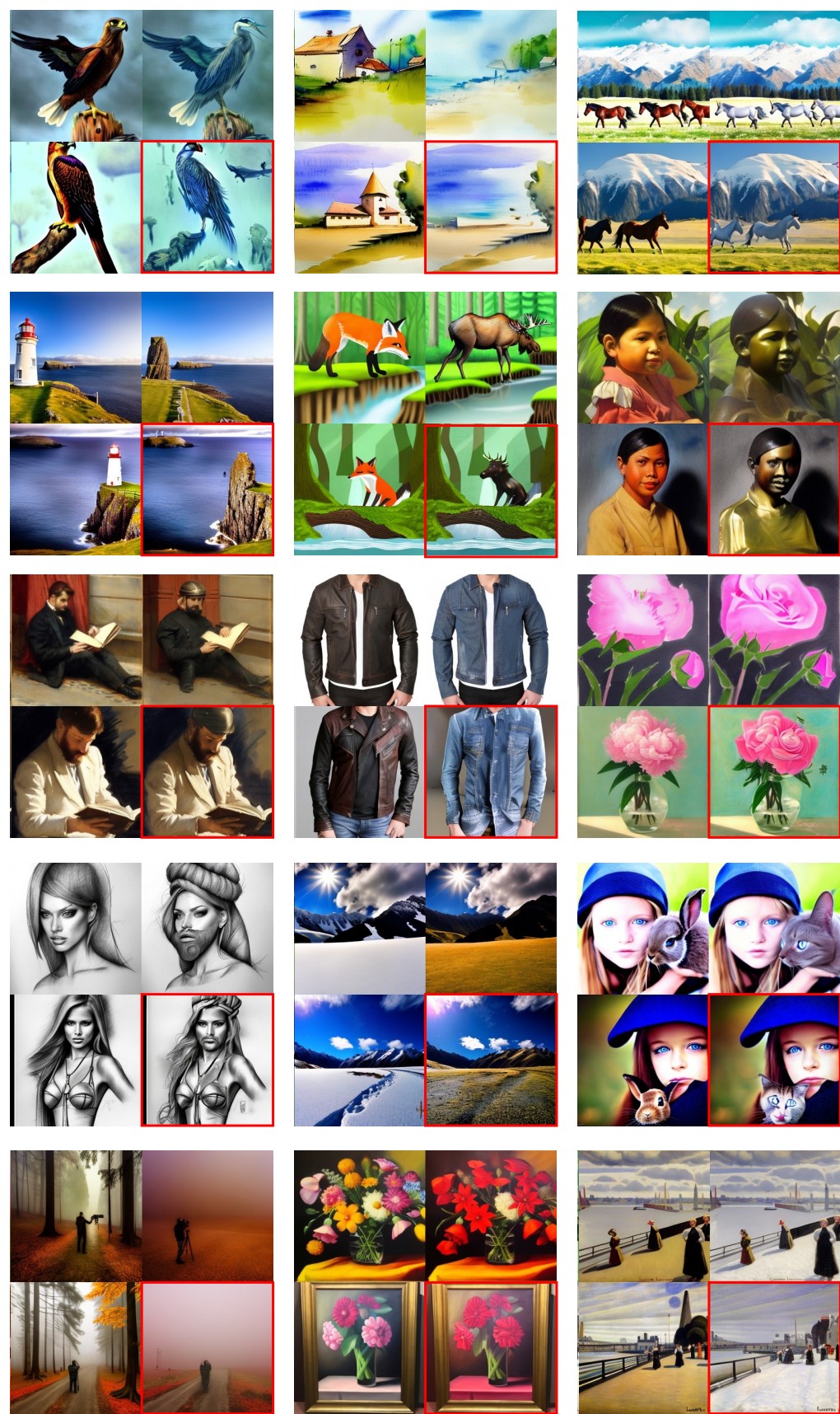

Figure 5: **Image Manipulation Results by Visual Instruction.**

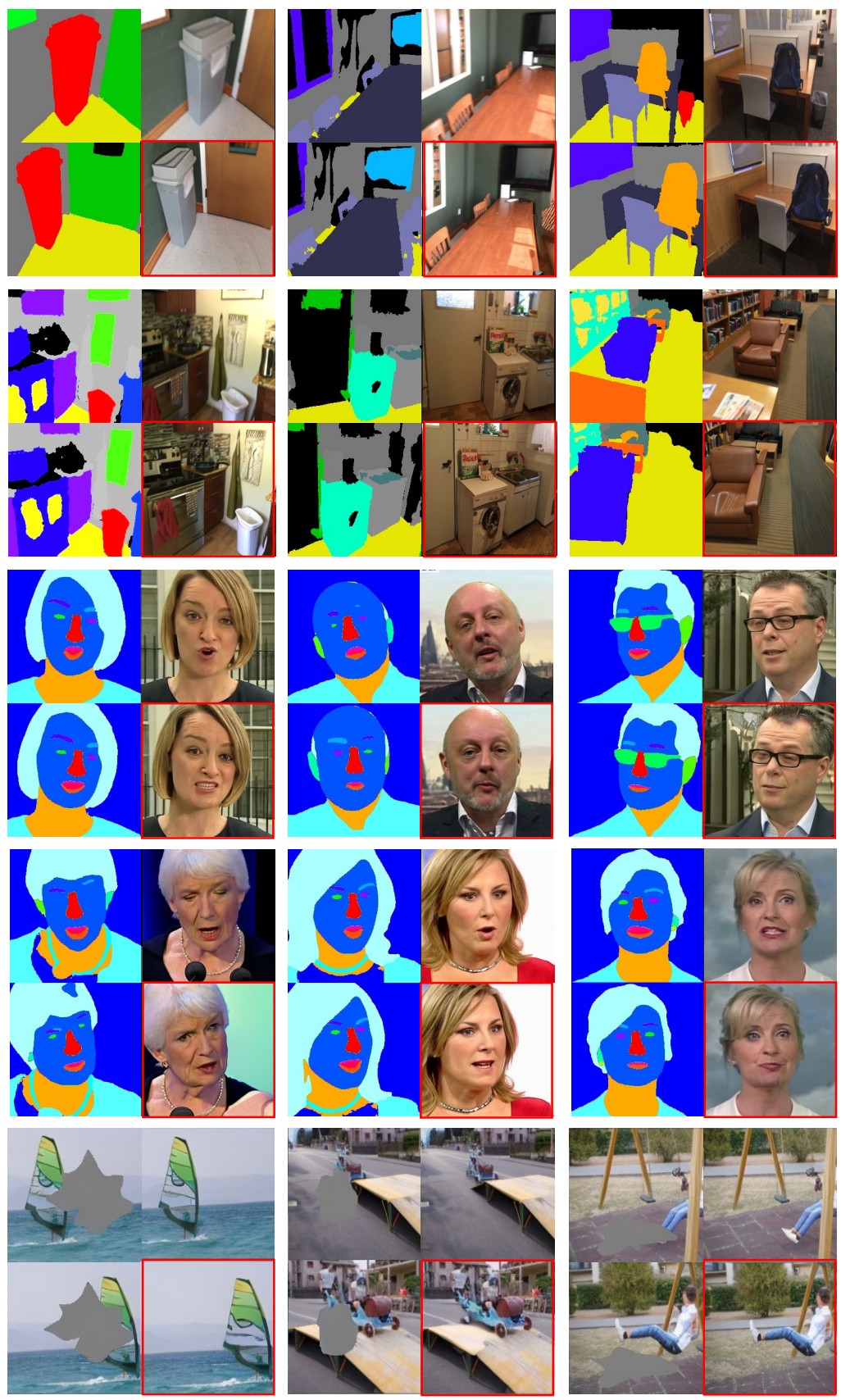

Figure 6: **Instruction Results on In-the-wild Dataset.**

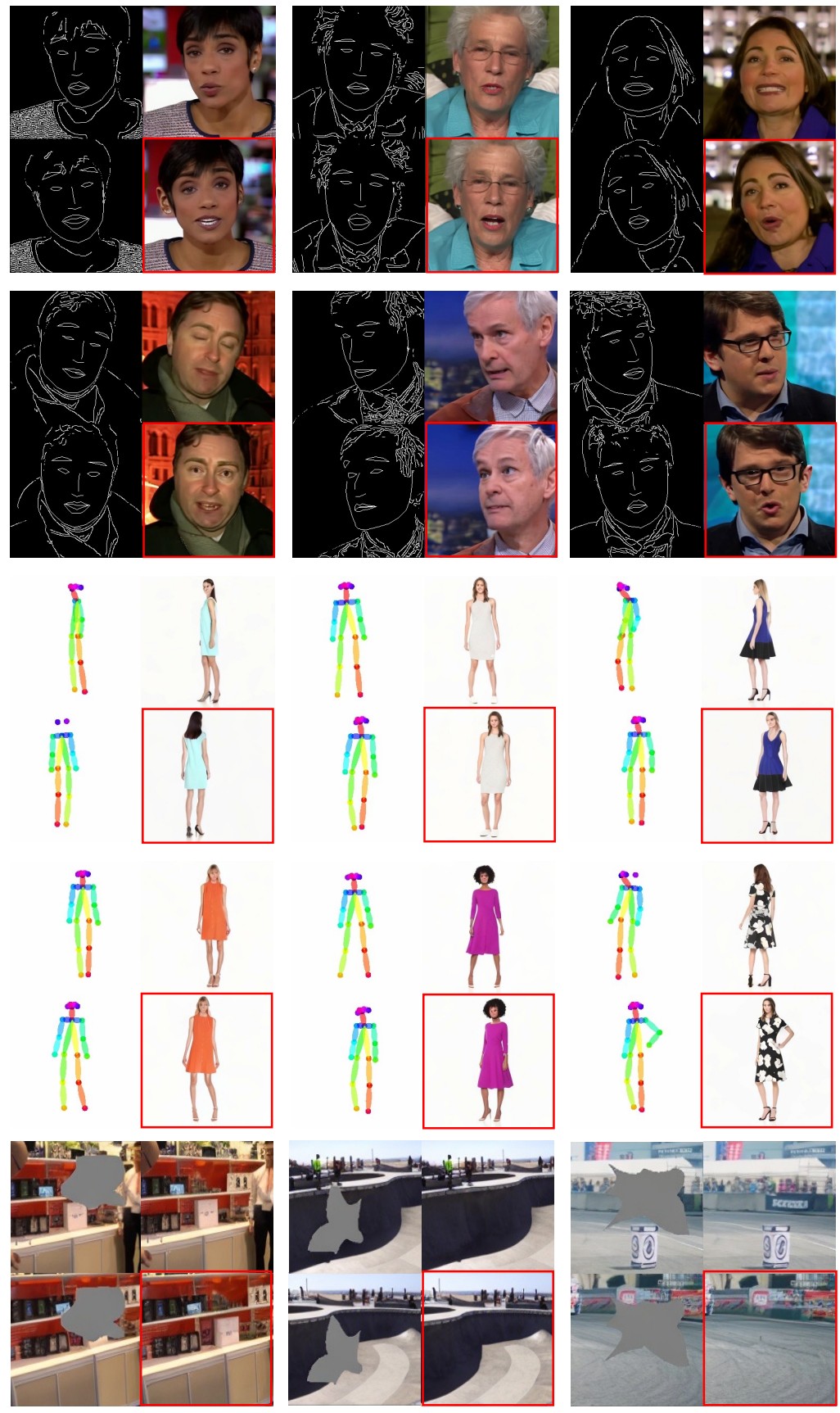

Figure 7: **Instruction Results on In-the-wild Dataset.**