# OpenReview forum: "ImageBrush: Learning Visual In-Context Instructions for Exemplar-Based Image Manipulation"
_NeurIPS.cc/2023/Conference — NeurIPS 2023 poster_

### Official Review · Reviewer_fTo3 · 2023-06-29

**Soundness:** 2 fair
**Presentation:** 3 good
**Contribution:** 2 fair
**Rating:** 5
**Confidence:** 4

**Summary:**

This paper proposed a novel scheme for image editing via exemplar-based.
An image edit will be represented as an in-context instructions.
To that end, the novel framework is trained to perform image editing as image in-painting task.
Experiment results indicate the advantages of this scheme over typical text-guided image editing frameworks.

**Strengths:**

**Originality/Novelty**:
- At a high level, this paper proposes a novel protocol for image editing: visual prompting. I believe this is an area will attract interests of both academic and applications.
- The designed framework, called ImageBrush, enable performing image manipulation in-context learning approach. Moreover, the interface design caters to user interest, which, although an add-on feature, proves beneficial for end-users.
- Through experiments and visual example, the paper demonstrates the merits of proposed protocol, which can be more acurate and efficient for image editing tasks.
- The authors did give efforts to try on multiple types of manipulation (e.g., seketon to fashion pose), instead of only standard image editing task.

**Presentation/ Writing**:
- Overall, the paper is well-organized and easy to follow.
- Framework illustration is fairly good; qualitative examples depicts the merits of the method.

**Weaknesses:**

**1. Some parts need to be revised to avoid over-claiming/ clearer**

1.1 The contribution mentioned in lines 72-73 overlaps with the Visual Prompting approach proposed in [1].
While I understand that this paper focuses on more specific tasks such as image editing and pose transfer, it would be advisable to use a more restrained tone.
Instead of making strong claims about the original idea of visual prompting, the authors could emphasize the concepts of "progressive inpainting" or "prompting for additional tasks."

1.2 The "Related Work" section should include references to "Visual Prompting" or "In-context learning" to provide more complete survey of prior work.

1.3. Figure 3 is unclear and requires further clarification. Please explain this Figure in the rebuttal.

**2. Flaws in experiments**

2.1. While the main contribution of this paper is an in-context learning scheme for image manipulation, I do not see any comparisons against existing in-context learning scheme (e.g., Visual Prompting [1] or Painter [2]).

I am not against the fact that the authors demonstrate the effective against "state-of-the-art" methods for each specific tasks (e.g.: compare with InstructPix2Pix in image editing).
I believe that it's crucial to compare against in-context learning methods as well.
For example: Authors should compare image in-painting task with Visual Prompting [1].

2.2. Lack of results for Interface design.
While I second this contribution, there is no qualitative results for this module in the main paper.
I have checked the Supplementary, only one example (edit of "putting a crown on the woman" is included). This example is not convincing itself, as I see the the crown in both approach is still not visually similar to the given crown in visual prompting.

Additionally, I feel like these results can be generated by different noises. For example, given the same visual prompt (e.g., "Put a crown on the woman", different noises can also lead to different visually different crown). For that reason, while I second this module, I am not convinced that this module works in practice.

2.3. Unfair comparison in video inpainting task.
Following [4] Video-FID (together with PSNR and SSIM) is used to evaluate the methods. Is it better to use Video-FID instead of FID?

**3. Miscellaneous**

3.1 Line 124- 138: While cite Latent Diffusion model [5], the formula are in pixel-space. Should the citation update to DDPM instead to suit the following formular?

3.2 Line 243: Missing citation for the "CLIP Direction Similarity" mentioned.

3.3 Line 218: Is the dataset name correct? My understanding is the authors indeed use the [InstructPix2Pix's CLIP-filtered-dataset]((https://github.com/timothybrooks/instruct-pix2pix#generated-dataset). There might be a misunderstood, as in InstructPix2Pix, "LAION Improved Aesthetics 6.5+" is used as just a small part of generating the final dataset (as written in [this](https://github.com/timothybrooks/instruct-pix2pix#11-manually-write-a-dataset-of-instructions-and-captions)).

**References**

[1] Amir et al., *Visual Prompting via Image Inpainting*, NeurIPS 2022.

[2] Xinlong et al., *A Generalist Painter for In-Context Visual Learning*, CVPR 2023.

[3] InstructPix2Pix et al., *InstructPix2Pix: Learning to Follow Image Editing Instructions*, CVPR 2023.

[4] Xueyan et al., *Progressive Temporal Feature Alignment Network for Video Inpainting*, CVPR 2021.

[5] Robin et al., *High-Resolution Image Synthesis with Latent Diffusion Models*, CVPR 2022.

**Questions:**

My initial rating is "Borderline accept" as I believe that while this paper is well-motivated, there are several revisions that need to be made.

In addition to the questions raised in the Weakness section, I also have some technical inquiries:

- Is the output size of all images 256x256? (which is smaller than other methods typically using 512x512).
- Was the model fine-tuned from Latent Diffusion, or trained from scratch?

**Limitations:**

The author has briefly addressed limitation. As a minor note, I recommend authors to include some qualitative results for limitation or challenging cases to encourage future research.

I strongly urge the authors to consider two key improvements during the revision process:

(1) If the authors intend to concentrate on image editing, it would be advisable to narrow down their focus to a single specific task. This would enable a more comprehensive exploration of the chosen task and provide greater clarity to the readers.

(2) If the authors aim to emphasize the modality aspect, I recommend conducting a more extensive comparison with existing works. This would help establish the unique contributions of the proposed method and demonstrate its superiority over previous approaches.

Regardless of the chosen direction, I believe that implementing these revisions would significantly enhance the paper's quality.

---

**After Rebuttal**:

I have read authors' rebuttal and other reviews. I thank authors for efforts to further address my concerns.

After careful consideration, I decided to maintain my rating as 5-Borderline accept.

My justification for not higher rating are:

1. The contribution about User's Interface (hybrid context injection strategy) is not (yet) fully validated.
1. Overlapping claims with Visual Prompting [1] paper (Also raised by [Reviewer zhtQ](https://openreview.net/forum?id=EmOIP3t9nk&noteId=d3daWKNPI3))
1. Some Figure, Table need clearer explanation or captions (Also raised by [Reviewer RQEf](https://openreview.net/forum?id=EmOIP3t9nk&noteId=SRf3GZGiV2))

---

> ### Author Rebuttal · Authors · 2023-08-09
>
> Thank you for your thoughtful comments.
>
> **_1. Some parts need to be revised to avoid over-claiming/ clearer_**
>
> 1.1-1.2. We greatly appreciate the constructive advice provided on our writing. To ensure accuracy and clarity, we will carefully tune down the claim and provide a more comprehensive introduction to the "In-Context Learning" work in our future version.
>
> 1.3. We compare our method with InstructPix2pix from two perspectives: aligning generated results with instructions (higher CLIP direction similarity) and retaining query image identity (higher image similarity). However, these two metrics can conflict to some extent, as not editing the query image would lead to perfect image similarity. Therefore, we separately measure them while controlling another variable, which involves averaging CLIP direction similarity for samples within the same CLIP image similarity bin and vice versa.
>
>
> **_2. Flaws in experiments_**
>
> 2.1 We further include additional comparison results against VisualPrompt in our appended PDF document. Fig.4 illustrates the qualitative results, revealing that ImageBrush yields plausible and realistic edits whereas VisualPrompt tends to produce conservative predictions and struggles on instruction comprehension. We believe this advantage can be attributed to the progressive synthesis nature of Diffusion. Moreover, quantitative results in Table.1 demonstrate the superiority of our approach against VisualPrompt on most tasks.
>
> 2.2 We have additionally incorporated more qualitative examples of our interface design in the attached PDF file. Fig. 1 depicts diverse examples with distinct emphasis on visual instructions. For example, synthesis of additional beards when the beard area is emphasized, the generation of more detailed hats when a bounding box is applied to the hat part, and the improved ability of our model to understand and execute the intention of editing a bucket to black when the emphasis is on the bucket. Furthermore, when a small object is added, our model successfully comprehends and synthesizes an object with the assistance of this input. Similarly, by placing emphasis on the background, more trees are synthesized in that region.
>
> 2.3 We provide further comparisons utilizing standard metrics, including PSNR, SSIM, and LPIPS, for the video inpainting task, as displayed in Table 1. Across these pixel-aligned metrics, TSAM demonstrates superior performance. This result can be attributed to the tendency of diffusion-based methods to over-imagine details. However, a higher FID value in our method indicates superior photorealism and a more pleasing visual quality.
>
> **_3. Miscellaneous_**
>
> Thank you for bringing this to our attention. We will ensure the citation is updated to DDPM for accuracy. Additionally, we will incorporate the proper citation for "CLIP Direction Similarity". As for the dataset name, we confirm that we utilize InstructPix2Pix's CLIP-filtered dataset, and we will proceed to modify it to ensure consistency.
>
> **_4. Questions_**
>
> 4.1 Regarding the image resolution, all images are of size 256x256, which is smaller than the typical size of 512x512. This is because our model relies on concatenating images to learn low-level appearance correlation via the self-attention mechanism, resulting in additional spatial dimension consumption.
>
> 4.2 We initialize our model with the latent diffusion weights.

---

> > ### Comment · Reviewer_fTo3 · 2023-08-10
> >
> > I have read your rebuttal.
> >
> > Thank you for your efforts in conducting additional comparisons to Visual Prompting [1].
> > However, I still have several concerns.
> >
> > **(1) Unfair comparison to Visual Prompting [1]**
> >
> > This question is also raised by Reviewer [#zhtQ](https://openreview.net/forum?id=EmOIP3t9nk&noteId=d3daWKNPI3).
> >
> > The authors have added additional quantitative and qualitative comparisons to Visual Prompting (as in the Rebuttal PDF file). However, I am doubtful about an unfair comparison.
> >
> > As far as I understand (maybe the authors can also double-check), *Visual Prompting [1] was not trained on any "in-the-wild" datasets* (e.g., ScanNet [2], UBC-Fashion [3], and Image Editing [4] datasets).
> >
> > Would it be unfair to Visual Prompting, as it was not trained for the Image Editing task?
> > (Figure 4, Rebuttal file).
> >
> > Moreover, I was surprised that Visual Prompting [1] (without being trained on any new tasks) performed reasonably well on most tasks compared to ImageBrush (e.g., on the Scannet dataset - PSNR: ImageBrush: 30.38 > Visual Prompting 29.34; on Inpainting - LPIPS: ImageBrush 0.816 > Visual Prompting: 0.787) (Table 1, Rebuttal file).
> >
> > These numbers are so close that it raises my concern.
> >
> > **If we train Visual Prompting [1] on the same dataset that was used to train ImageBrush, would Visual Prompting achieve a similar level of performance, or possibly even surpass ImageBrush?**
> >
> > (If the answer is yes... *Then what is the advantage of ImageBrush except from training on a larger datasets?*
> >
> > It is also worth to note that ImageBrush was finetuned from Latent Diffusion, which was trained on billions examples of [LAION dataset](https://laion.ai/blog/laion-5b/))
> >
> > **(2) Figure 3 is still unclear to me**.
> >
> > I don't understand the quantity of the red and green columns.
> >
> > Is that the number of images that match the specific threshold on the x-axis or y-axis?
> >
> > How many images are used to plot this Figure? Where do these images come from? Are these images used in training?
> >
> > **(3) Doubt about Interface design.**
> >
> > I am still left with questions about Figure 1 in the Rebuttal PDF file.
> >
> > 3.1. What's the green-like bounding box in every example image?
> >
> > 3.2. The authors still haven't answered my question: "I feel like these results can be generated by different noises. For example, given the same visual prompt (e.g., "Put a crown on the woman"), **different noises can also lead to different visually different crowns.**" Is this statement true?
> >
> > To be specific, using the same example-pair and query image, is it true that I can generate different outputs by applying different noises? In other words, can different noises lead to different images, each focusing on a different aspect (e.g., one image focusing on the hat, other images focusing on the beard)?
> >
> > Please refer to [Figure 12 in the InstructPix2Pix paper](https://arxiv.org/abs/2211.09800) for a visual example of what I am trying to depict.
> >
> > My other unanswered questions are as follows:
> >
> > **(4) Unfair comparison in video inpainting task**.
> >
> > "Following [4], Video-FID [...] is used to evaluate the methods. Would it be better to use Video-FID instead of FID?"
> >
> > **(5) Questions about the focusing goal of this paper (Stated in Limitations)**.
> >
> > I want to know how the authors identify this work. Will it be a novel method for image editing, or will it be a powerful modality framework (e.g., capable of solving a wide range of tasks)?
> >
> > Thank you!
> >
> > *Reference:*
> >
> > [1] Amir et al., *Visual Prompting via Image Inpainting*, NeurIPS 2022.
> >
> > [2] Angela et al., *ScanNet: Richly-annotated 3D Reconstructions of Indoor Scenes*, CVPR 2017
> >
> > [3] Polina et al., DwNet: *Dense warp-based network for pose-guided human video generation*, BMVC 2019
> >
> > [4] Tim et al., *InstructPix2Pix: Learning to Follow Image Editing Instructions*, CVPR 2023.

---

> > > ### Author Response · Authors · 2023-08-12
> > >
> > > **_Unfair comparison to Visual Prompting [1]._**
> > >
> > > 1) For fair comparison we have carefully re-trained [1] using the same dataset we employed for ImageBrush. Without training on the same dataset, their performance is super terrible, which is uninformative and thus not shown.
> > >
> > > 2) Actually, both Table 1 and Figure 4 illustrate their performance after training on the same tasks as ours. I guess that’s why it feels reasonably well. From Table 1, we could see that even training on larger datasets as ours, their results are still below ImageBrush. Additionally, Figure 4 reveals that they tend to produce conservative results and lack details. Although it is a good strategy to attain decent metric value, their results are not photorealistic, which will largely limit the real-world application.
> > >
> > >
> > > **_Clarification of Fig. 3_**
> > >
> > > 1. The quantity of green and red columns does not correspond to the count of images. Its height reflects the average value (CLIP Direction/Image Similarity) derived from all the images encompassed within the respective bins.
> > >
> > > 2. As for the bins, consider the left figure as an illustration. For the first green and red columns sitting around 0.8, we compute the mean CLIP Direction Similarity for all images falling within the CLIP Image Similarity range of $[0.8-1/6, 0.8+1/6]$.
> > >
> > > 3. To draw this figure, we employ a dataset of 10k images that remained unseen throughout the training phase, as stated in the Dataset of Section 4.1.
> > >
> > >
> > > **_Doubt about interface design._**
> > >
> > > 1. We apologize for the oversight in visualization. It appears that we inadvertently depict the bounding boxes produced by Ground DINO.
> > >
> > > 2. After a close examination of Figure 12, we concur that diverse noise sources can indeed yield varied outcomes due to the intrinsic nature of DDPM. But utilization of  bounding boxes could further steer the model's attention towards the highlighted area. This effect is naturally facilitated by the statistical correlation between bounding boxes and relevant editing regions. To elaborate, Ground DINO enables the automatic identification of the most pertinent spatial region within the image. This explicit indication of the editing intention furnishes our model with an interface to more effectively capture such intentions. Our statistical observations and qualitative examples of Rebuttal file validate its proficiency in capturing intention.
> > >
> > > 3. Regarding the interplay between bbox and noise effects, we recognize that this deserves further exploration. In future, we will show more examples with less ambiguity to demonstrate its efficacy. We really appreciate your constructive insight.
> > >
> > >
> > >
> > > **_Unfair comparison in video inpainting task._**
> > >
> > > 1. For video inpainting tasks, the Video-FID stands as a crucial metric to evaluate temporal aspects of video synthesis. Here we include a comparison with the strong baseline TSAM(ImageBrush 175.25 vs. TSAM 158.30). ImageBrush achieves close performance even without special designs to harness temporal information.
> > >
> > > 2. Notably, the computation of the Video-FID relies on the i3d feature extractor, which has been pre-trained on an extensive collection of videos. Consequently, the use of smoothed video features is favored. But for our approach, which lacks temporal awareness, the extracted features are likely to diverge significantly from their distribution. As for the FID metric that evaluates each frame independently, the features extracted by the Inception model tend to align closely with the Ground Truth. This discrepancy in feature behavior explains why our model attains a higher FID score while yielding a comparatively lower Video-FID score.
> > >
> > >
> > > **_Questions about the focusing goal._**
> > >
> > > 1. Characterizing this work as a novel image editing approach may not be entirely accurate. The selection of image editing is rooted in the fact that image manipulation contains abundant contexts, offering potential solutions for practical real-world applications.
> > >
> > > 2. We prefer to locate our work as a visual in-context learning investigation from the perspective of image generation, which is complementary to VisualPrompting. While they show potential in visual in-context learning, their focus remains on recognition like detection and segmentation, thereby having limited applications. But directly extending their method to generative tasks will encounter challenges on capturing editing intention and produces conservative synthesis, as Fig.4. Therefore, we introduce the diffusion mechanism, which guarantees intention comprehension and realistic synthesis. Our work incorporates a meticulously designed prompt encoder to grasp the underlying manipulation intention and an interface to enhance its understanding. In this way, we open avenues for novel applications (see Common Response). This extension significantly expands the range of potential applications and signifies a substantial step towards establishing a foundational model for vision-based applications.

---

> > > > ### Comment · Reviewer_fTo3 · 2023-08-14
> > > >
> > > > Thank you for providing clarification. All my concerns about unfair comparisons have been resolved.
> > > >
> > > > I understand Figure 3 now, but I'd like to offer a minor note. Since I didn't understand the concept of Figure 3 without your explanation, it might be possible that readers find it hard to understand too.
> > > > My suggestion would be a histogram-like graph (e.g., [Figure 7 in Visual Instruction Inversion paper](https://arxiv.org/pdf/2307.14331.pdf)), or a line graph (e.g., [Figure 8 of InstructPix2Pix](https://arxiv.org/pdf/2211.09800.pdf)), for a clearer interpretation. However, the choice of the final plot style is up to authors.
> > > >
> > > > It's unfortunate that the Interface module requires further thorough investigation. I second the potential of this module, as it could open the way for a novel interactive system for image editing.
> > > >
> > > > Finally, I (personally) share the same view as the authors regarding the focus of the goal of this work: visual in-context learning.
> > > >
> > > > ---
> > > >
> > > > I will read other review again and finalize my rating. Thank you!

---

> > > > > ### Author Response · Authors · 2023-08-18
> > > > >
> > > > > Thank you for your constructive suggestions, which have been very helpful to us. We appreciate your understanding and recognition.
> > > > >
> > > > > Indeed, we have realized the confusion in Figure 3, and we plan to reconstruct it by referring to Figure 7 in the [Visual Instruction Inversion](https://arxiv.org/pdf/2307.14331.pdf) [1] paper, making it easier to understand. We will also revise the description of our motivation to have a clearer focus and include more validation results for the Interface module in subsequent versions.
> > > > >
> > > > > As for the novel interactive system for image editing, we plan to open-source our demo and code in the future, allowing more users to participate and try it out. This is crucial for iterating and exploring a more user-friendly image editing application paradigm.
> > > > >
> > > > > In regards to visual in-context learning, we hope that our work can inspire more follow-up studies to explore new ways of using visual generative models and more general application scenarios.
> > > > >
> > > > > *Reference:*
> > > > >
> > > > > [1] Nguyen et al., *Visual Instruction Inversion: Image Editing via Visual Prompting.* arXiv preprint arXiv:2307.14331.

---

### Official Review · Reviewer_uVk2 · 2023-07-09

**Soundness:** 3 good
**Presentation:** 3 good
**Contribution:** 3 good
**Rating:** 6
**Confidence:** 2

**Summary:**

The authors propose a novel, visual-based method for image editing, using a pair of example images as instructions for users to specify the desired operations. The proposed idea uses the details and nuance in the example pairs to accomplish complex editing goals. The authors formulate this approach as a diffusion inpainting task, where the instruction pair and the query image are given and tiled as a grid. To achieve the desired quality, the visual prompt and region of user interest are injected into the diffusion structure as conditions, and the whole pipeline is trained as a latent diffusion model with classifier-free guidance. The authors also developed a metric to assess the quality of manipulations performed and conducted quantitative and qualitative studies.

**Strengths:**

1. The proposed method demonstrates a new way to manipulate images using visual instructions only.
2. The model generalizes well on downstream tasks compared to other methods requiring task-specif models.
3. The authors propose a new metric to evaluate image manipulation tasks.

**Weaknesses:**

1. For many common image editing tasks, it will be challenging or tedious for users to find the example pairs. For example, if a user wants to convert a photo into a painting by a specific artist, preparing the paired picture for the given painting will be hard without using existing pre-trained style-transfer models.

2. The authors think one of the disadvantages of using text prompts is that text prompts are abstract. However, the proposed method is also abstract if users want specific, exact manipulations. For example, if we're going to make the spout of a teapot 2 cm longer or want a particular proportion of the teapot shape, finding the pair examples and ensuring the model does the exact manipulations will be challenging.

3. Conducting user studies to support the assumptions would be great. For example, is it challenging for the users to find the pairs for their desired editing? Do the results look better to them (FID and CLIP scores are shown not to reflect human preferences/aesthetics in prior works like [1])?

[1] Better Aligning Text-to-Image Models with Human Preference by Wu et al.

**Questions:**

1. It is unclear how the image grid, Xt, is represented. Figure 2 shows that only I' is converted to noise after the forward diffusion process. However, in LDMs, the diffusion process is in the z space, where the whole image is encoded.

2. Is resolution one of the limitations of this method since the inpainting area is only 1/4 of the input image grid?

3. How the model is trained using the four datasets mentioned in 4.1 is unclear. Is it trained from scratch?

4. I am curious about how necessary the inpainting setup is. For example, what will the model perform if the task is to generate only I' instead of the whole grid (in other words, the visual clues and instructions are comping from cross-attention only)?

5. Have the authors tried incorporating the region of interest using different methods (such as masking in latent space similar to eDiff-I)?

**Limitations:**

Yes, the limitations are addressed.

---

> ### Author Rebuttal · Authors · 2023-08-09
>
> Thank you for your thoughtful comments.
>
> **_Difficulty in finding example pairs._**
>
> As stated in the common response, we highlight that the practical application of our proposed system is to offer an efficient tool for users to automatically streamline their desired editing operation to a new-coming image, which acts like the Format Painter in Office Software, rather than targeting editing a specific image. Specifically, it is natural and intuitive to instruct an image with language considering its easy access. But when it comes to a situation where users want to apply analogous manipulation referring to his or others’ finished edits. In this case, it is particularly challenging to tackle with text condition or single image conditioned diffusion models because it requires inducing the underlying operations and imposing the intended manipulation on the target image. Furthermore, since the proposed system behaves like a ImageBrush, users are capable of accomplishing this solely with a pair of images without recording tedious settings such as a series of parameters in Photoshop.
>
> To sum it up, users could always manually create or find their preferred visual instructions, with which our model enables them to conveniently leverage the implicit operations. Admittedly speaking, we agree that this framework will be more user-friendly if it could support visual instructions without relying on spatial alignment of the example image. Regarding how to deal with more versatile visual prompts, we will leave it for future research.
>
>
> **_Challenges in Precise Manipulations._**
>
> We value the insightful observation made by the reviewer. Indeed, finely manipulating the shape of a teapot presents challenges for both language-based and visual instructions. However, in situations involving unarticulated object appearances and artistic styles, our proposed system exhibits advantages over text-based guidance. Moreover, our model's capabilities enable novel applications as elaborated in the last section.
>
> For scenarios where users require exact or specific manipulations, such as extending the spout of a teapot by 2 cm or adjusting the proportions of the teapot shape, it might be beneficial to combine both visual instructions and text prompts to guide the image manipulation process. In this case, the visual instruction can provide the overall style or appearance information, while the text prompt can convey the specific details or numeric parameters. Integrating both modalities can potentially improve the accuracy and precision of the manipulation while maintaining the advantages of visual instructions in capturing complex concepts.
>
> Furthermore, we acknowledge that finding suitable pair examples for highly specific manipulations can be challenging. One potential solution to this issue is to develop an interactive system that allows users to provide iterative feedback or refine the visual instructions during the manipulation process, which could help the model better understand and satisfy the user's requirements.
>
> **_Conducting User Studies._**
>
> Thank you for your suggestion. We will consider incorporating this type of user study as part of our future work, as well as addressing the limitations of metrics such as CLIP scores and FID in our evaluation.
>
>
> **_Unclear about diffusion process._**
>
> In LDMs, the entire z space is subject to the addition of noise. In the reverse process at each timestep, we re-inject clean $\textbf{E}$, $\textbf{E}'$, and $\textbf{I}$ into the UNet to provide a clean context, allowing the noise prediction task to focus on recovering the inpainting task of $\textbf{I}'$.
>
> **_Unclear about image grid and resolution problems._**
>
> Currently, this limitation exists wherein we can generate results only at a resolution of 256x256, despite the model's inherent capacity to produce outputs at 512x512.  We remain optimistic about overcoming this constraint in the future. One potential solution could involve concatenating the inputs channel-wise, a strategy that would preserve resolution without compromise.
>
>
> **_Is it trained from scratch?_**
>
>  We initialize our model with the latent diffusion weights.
>
>
> **_How necessary the inpainting setup is._**
>
> Our primary motivation for constructing the inpainting task is to inject low-level features from $\textbf{E}$, $\textbf{E}'$, and $\textbf{I}$ through the self-attention branch rather than relying solely on cross-attention. If we do not construct this inpainting task and solely incorporate features into the UNet's cross-attention to control diffusion, the image features would lose a significant amount of detail after being encoded by CLIP. Moreover, since the injection takes place in the latent space of LDMs, low-level information must be restored using the knowledge of the VAE's encoder.
>
> By using the grid-like image inpainting approach, features of $\textbf{E}$, $\textbf{E}'$, and $\textbf{I}$ are fully aligned with the predicted image $\textbf{I}'$, as they all pass through the VAE's encoder. This method allows us to retain as much low-level information as possible in a simple and effective manner. If the loss of resolution is a concern, we can adopt a channel concatenation approach (however, this would require allocating separate parameters for each image to avoid alignment issues).
>
> **_Incorporating the region of interest using different methods._**
>
> While we have not explicitly experimented with this method in our current work, it could be an interesting direction for future research. Masking in the latent space or re-weight the attention matrix may potentially lead to further improvements in image generation by more accurately preserving the user's region of interest and better aligning with the provided instructions. We appreciate the reviewer's suggestion and will try incorporating those approaches in future studies to assess their efficacy.

---

> > ### Comment · Reviewer_uVk2 · 2023-08-21
> >
> > Thank the authors for the rebuttal. Based on our discussion above, I would suggest the authors to clarify the limitation and use cases of the proposed method in the revision. I will raise my score to 6.

---

### Official Review · Reviewer_h3g5 · 2023-07-10

**Soundness:** 3 good
**Presentation:** 4 excellent
**Contribution:** 2 fair
**Rating:** 4
**Confidence:** 5

**Summary:**

This paper introduces a novel image manipulation framework called ImageBrush, which allows for accurate and efficient image manipulation without the need for complex language descriptions. The framework consists of a diffusion-based generative model coupled with a hybrid context injection strategy, which facilitates better correlation reasoning. The authors also introduce a bounding box incorporation module to take user interactivity into consideration. The framework is evaluated on various downstream tasks and exhibits robust generalization abilities. Overall, the results demonstrate that ImageBrush generates compelling manipulation results aligned with human intent and paves the way for future vision foundation models.

**Strengths:**

- The paper introduces a new image manipulation framework called ImageBrush, that does not rely on complex language descriptions. This may help mitigate severe prompt engineering efforts in applications.
-  A diffusion-based generative model coupled with a hybrid context injection strategy is proposed, which is a technically rigorous approach to image manipulation.
- Extensive experiments are conducted on various downstream tasks and discuss this work's generalization abilities.
- The paper is well-written and easy to follow, with clear explanations of the technical concepts and experimental results.

**Weaknesses:**

- Although this work utilizes an in-context learning strategy to avoid heavy prompt engineering effort, the image prompt itself seems to arise another challenge. Take Fig. 4 as an example, the provided instruction (tiger pairs) itself necessitates heavy editing skills, that require another expert (either human or modern algorithms) to provide a snowy tiger. I'm worrying that this will not be practical in real scenarios given users mostly can not provide such a good pair of instructions, and writing prompts would be relatively easier. A more practical way is to provide a tiger with another snowy example (not have to be a tiger) so that it will be easier to collect inputs. However, this paper does not show this demo.

- Besides the input collecting issue, it seems that the instruction and the input image are very similar in all provided results. If that is the requirement of making the proposed algorithm succeed, it will be further difficult to be applied to real scenarios, since one can not pre-collect corresponding instructions due to the in-the-wild testing case is not predictable.

- The authors propose a new evaluation metric in Sec 4.4. I agree that instruction-based editing needs a reliable quantitative metric to help justify the performance of each algorithm, but I doubt whether it is a good idea to put it as one of the contributions. Proposing a new metric normally requires a large amount of user study that can faithfully provide evidence to show that the new metric is aligned with human perception, which is lacking in this work. I understand that will cost a huge effort and would probably produce a completely new submission. Therefore, I would suggest the author not claim this as a "novel" finding and lower its importance in the manuscript.

**Questions:**

N/A

**Limitations:**

See weaknesses.

---

> ### Author Rebuttal · Authors · 2023-08-09
>
> Thank you for your thoughtful comments.
>
> **_Visual instruction collection seems a challenge._**
>
> As stated in the common response, we highlight that the practical application of our proposed system is to offer an efficient tool for users to _automatically streamline their desired editing operation to a new-coming image_, which acts like the Format Painter in Office Software, rather than _targeting editing a specific image_. Specifically, it is natural and intuitive to instruct an image with language considering its easy access. But when it comes to a situation where users want to apply analogous manipulation referring to his or others’ finished edits. In this case, it is particularly challenging to tackle with text condition or single image conditioned diffusion models because it requires inducing the underlying operations and imposing the intended manipulation on the target image. Furthermore, since the proposed system behaves like a ImageBrush, users are capable of accomplishing this solely with a pair of images without recording tedious settings such as a series of parameters in Photoshop.
>
> To sum it up, users could always manually create or find their preferred visual instructions, with which our model enables them to conveniently leverage the implicit operations. Admittedly speaking, we agree that this framework will be more user-friendly if it could support visual instructions without relying on spatial alignment of the example image. At the same time, it’s quite an intriguing idea to obtain a snowy tiger when prompting for tiger and snow, which shares a similar setting with CoDI. However, this type of prompting is a little bit different from our work where we rely on paired manipulated images to indicate the desired editing operation. Regarding how to leverage these soft visual prompts, we will leave it for future work.
>
>
> **_Similarity in instruction and query image constraints real-world scenario applications._**
>
> As depicted in the accompanying PDF document, Figure 2 demonstrates our method's versatility in image editing across a diverse range of query inputs, which suggests our model's capability to handle a wide spectrum of query scenarios. Additionally, Figure 3 serves to illustrate our framework's adaptability – it not only accommodates in-the-wild images but also, to a certain extent, grasps previously unseen visual instructions.
>
> Consequently, once a user prepares an example alongside its corresponding retouching instance, they can efficiently and effortlessly apply this operation to a similar collection of images. By grasping the inherent editing intention within the provided instruction, the proposed model becomes a valuable tool in executing analogous creative tasks on pertinent target images.
>
>
> **_Evaluation metric contribution._**
>
> We appreciate the suggestion to tune down the introduced evaluation metric in our manuscript. We acknowledge that proposing a new metric entails thorough user studies and substantial evidence to establish its alignment with human perception. In our study, the proposed evaluation metric solely serves as an initial endeavor to enhance the assessment of this novel generative task. We will proceed to revise our manuscript to more accurately convey the significance of the evaluation metric.

---

### Official Review · Reviewer_zhtQ · 2023-07-11

**Soundness:** 3 good
**Presentation:** 3 good
**Contribution:** 3 good
**Rating:** 7
**Confidence:** 5

**Summary:**

In this paper, the authors propose a new method that can perform visual prompting via a pair of exemplar images. The authors introduce and train a new model for this task and evaluate the model performance on various datasets, showing strong empirical performance and a wide range of applications. They also propose new evaluation metrics to evaluate their method.

**Strengths:**

This model introduced in this paper solves a very interesting and important application problem that is exemplar-based visual prompting of image generative models. The results look very promising and this model can be easily directly applied in a large range of intuitive applications by daily users.

**Weaknesses:**

1. This paper is very similar to the paper “Visual Prompting via Image Inpainting” (https://arxiv.org/pdf/2209.00647.pdf). However, the authors didn’t compare their method with this paper.
2. The quantitative evaluations are not very comprehensive. For example, for tasks like semantic segmentation masks to image, the authors could have reported some standardized metrics like IoU.
3. Similar to (2), quantitative analysis on user-specified regional editing can also be included by comparing visual distances such as LPIPS.

**Questions:**

1. How general is ImageBrush? How does the model perform on more OOD tasks/inputs (i.e. the prompts to which similar examples are not seen in training)? Or does it require training for each task?
2. How similar does the visual prompt and query image should look? Can the authors give some additional ablation study on the similarity between the visual prompt and thequery images?
3. Related to Weakness (1), can the authors compare their method with Visual Prompting via Image Inpainting?
4. Related to Weakness (2) and (3), can the authors report some standardized metrics for tasks like semantic segmentation mask to image, keypoint to image and user-specified regional editing?

~~I am happy to raise my score if the authors add experiments to address all my concerns.~~

***After the rebuttal discussion, I would like to raise my score from 5o to 7.***

**Limitations:**

Although presented in the paper, the discussion of limitations and potential ethical issues is very limited. I would encourage the authors to extend this discussion by providing some failure cases and suggesting some future plans for safeguards, etc.

---

> ### Author Rebuttal · Authors · 2023-08-09
>
> Thank you for your thoughtful comments.
>
> **_Weakness_**
>
> 1. We further include additional comparison results against VisualPrompt in our appended PDF document. Fig.4 illustrates the qualitative results, revealing that ImageBrush yields plausible and realistic edits whereas VisualPrompt tends to produce conservative predictions and struggles on instruction comprehension. We believe this advantage can be attributed to the progressive synthesis nature of Diffusion. Moreover, quantitative results in Table.1 demonstrate the superiority of our approach against VisualPrompt on most tasks.
>
> 2. The appended PDF document also includes a quantitative evaluation with standard metrics, such as PSNR, SSIM, and LPIPS. As depicted in Table 1, our approach demonstrates superior performance in terms of SSIM and LPIPS scores, highlighting its effectiveness in maintaining perceptual quality and preserving high-level structures.
>
> 3. Regarding the quantitative analysis of user-specified regional editing, we have also included the corresponding results in Table 1. Notably, the "ImageBrush-w-BB" model takes the region of interest (ROI) into consideration. However, we observe that there is no significant divergence in terms of LPIPS, PSNR, and SSIM metrics. This outcome could be attributed to the inherent diversity of the editing process, which might not always align precisely with a specific ground truth (GT). As a result, we have conducted an additional evaluation using CLIP direction similarity, revealing superior performance when enhanced with user-specified region injection (0.18 vs. 0.17). Furthermore, we incorporate more qualitative results to further demonstrate the effectiveness of this module in Fig. 1.
>
> **_Questions_**
>
> 1. To assess the model's generalizability, we have included extra in-the-wild examples in the PDF file. In the center of Figure 3, we showcase edited outcomes using real-world query images sourced from the Internet. Despite the distinct nature of these query images compared to the training distribution and example instructions, our model is capable of generating plausible results. On the right-hand side, we present out-of-distribution (OOD) cases. Notably, our model was not exposed to animal edges and keypoints, depth maps, or facial normals during the training phase. However, it demonstrates the ability to grasp the primary intent and produce reasonable edits. These examples highlight the model's capacity to learn fundamental visual correlations even in previously unseen scenarios to a certain extent.
>
> 2. Regarding the alignment between visual prompts and query images, we present a collection of qualitative examples encompassing diverse query contexts in the attached document (Fig. 2). The results suggest our model's competence in generating coherent and reliable outcomes across various query scenarios. Generally, a query image with a CLIP Similarity surpassing 0.7 tends to yield satisfactory performance. To analyze the relationship between CLIP Image Similarity and CLIP Direction Similarity, we calculate a correlation matrix and obtain a Pearson Correlation Coefficient of 0.195, accompanied by a T-Statistic of 10.58, implying that higher similarity leads to better performance to a certain extent.
> Furthermore, we present an additional ablation study concerning their relationship listed below. From the results we can see that, on the whole, editing performance demonstrates improvement as the query image aligns more closely with the example instructions.
>
>
> | Ablation Study|  |  |  |  |  |  |  |  |
> |---------------------------|------|------|------|------|------|------|------|------|
> | CLIP Image Similarity     | 0.76 | 0.79 | 0.82 | 0.85 | 0.88 | 0.91 | 0.94 | 0.97 |
> | CLIP Direction Similarity | 0.10 | 0.12 | 0.12 | 0.13 | 0.15 | 0.17 | 0.21 | 0.27 |
>
>
>
> 3. Addressed in Weakness 1.
>
> 4. Addressed in Weakness 2.
>
>
> **_Limitation_**
>
> Thank you for the valuable feedback. In our future revisions, we will certainly expand the limitations section as suggested. This expansion will encompass the visualization and analysis of failure cases. In our empirical observations, we've noted challenges when modifying small objects, possibly stemming from confusion with their semantic concept.
> As for the future work, our framework favors spatially aligned instruction examples because most  instances within our dataset are aligned. Therefore, our future endeavors will focus on integrating language cues to improve the comprehension of small objects, while also exploring visual instruction with non spatially aligned example images.

---

> > ### Comment · Reviewer_zhtQ · 2023-08-18
> > **Thank you for your response**
> >
> > Thank you for your responses to my concerns. I believe this rebuttal has sufficiently addressed all my concerns and therefore I would like to raise my score from 5 to 7.

---

### Official Review · Reviewer_RQEf · 2023-07-26

**Soundness:** 3 good
**Presentation:** 2 fair
**Contribution:** 3 good
**Rating:** 6
**Confidence:** 4

**Summary:**

The paper proposes a novel image manipulation technique based on diffusion and visual context images. The authors argue that context from language can only communicate abstract ideas while giving visual context can bring across inexpressible concepts like art style or complex appearances.

The novel technique adopts the 4-image-quadrant concept where the 4th quadrant is in-painted by to model to match some target (similar to SegGPT and Painter). But instead of in-painting segmentation masks, depth maps, or key-points in the 4th quadrant, they in-paint an edited version of the query based on transformation style or content given the context (prompts).

To improve the grounding process, this work proposes to add an additional prompt encoder branch of the architecture which is fed into the diffusion process via Cross-Attention. This branch also allows optional guidance through bounding boxes on which the in-painting process should add additional focus. Bounding boxes can either be provided by drawing them or via language by using Grounding Dino.

Finally, this work introduces a novel evaluation metric for exemplar-based image manipulation which does not need ground truth or human evaluation.

**Strengths:**

- The paper is well-written and mostly easy to read.
- The proposed approach is novel and the paper does a good job of highlighting the motivation for this line of work. The method is evaluated on multiple benchmarks and shows promising results. Comparisons are made with other image manipulation methods wherever possible.
- The work is visually attractive with many examples of generated images and comparisons with similar methods
- The authors highlight limitations of their approach by showcasing examples where the model works well but also cases where the model breaks down and does not work well.
- Authors show qualitative and quantitative results on multiple datasets, showcasing the superiority of their method.

**Weaknesses:**

- The description of the design of the prompt encoder is minimal and it would be interesting to have a more in-depth description. Specifically to how the different context image embeddings are combined. Are they just concatenated like the bounding box embeddings?
- The section on the novel evaluation metric is somewhat unclear, specifically why the two directions of evaluation/combination of query and prompts are chosen. (Maybe more descriptive variable naming could help)
- Generally, captions of figures and tables are minimal and contain little information. Specifically, figure 6 is confusing and could use an explanatory caption.
- Equations 4 to 6 are unclear and not understandable.
- Supplementary material is not formatted as an appendix.

**Questions:**

Would the architecture still work if you would drop the 4-quadrant concept for two quadrants and only injecting context/prompts through cross attention?

**Limitations:**

Authors adequately address limitations.

---

> ### Author Rebuttal · Authors · 2023-08-09
>
> Thank you for your thoughtful comments.
>
> **_How the different context image embeddings are combined._**
>
> We apologize for the lack of clarity in the description of the prompt encoder and will refine this section for improved clarity. Yes, the different context image embeddings are indeed concatenated, just like the bounding box embeddings. Throughout our investigation, we explored alternative strategies, such as leveraging differences in CLIP features between instruction images for information injection, as well as incorporating supplementary operations to characterize editing procedures. However, our experimentation indicated that these approaches failed to yield substantial advantages. It turns out the uncomplicated concatenation of embeddings already demonstrates efficacy in facilitating the model's comprehension of the editing intent derived from multiple images. Hence, we opted for straightforward concatenation for the sake of simplicity.
>
>
>
> **_The novel evaluation metric is somewhat unclear._**
>
> We apologize for any confusion that may have arisen in our explanation of the novel evaluation metric. We will revise those parts to provide greater clarity and incorporate descriptive variable names. The rationale behind the choice of the two directions of evaluation and the combination of query and prompts is to assess the fidelity of our model's image manipulation in terms of both following the provided instructions (prompt fidelity) and preserving the content of the input image (image fidelity).
>
> * Prompt Fidelity: This metric evaluates the model's ability to follow the provided instructions, which are given in the form of exemplar image pairs. By using the instructions formulated from the query image and generated image, we expect the model to manipulate one of the example images to match the other one. A high prompt fidelity score indicates that the model effectively captures the underlying instructions from the examples and successfully applies the desired transformation.
>
> * Image Fidelity: This metric evaluates the extent to which the model preserves the content of the input image when performing the manipulation. If the manipulated image maintains the content of the input image, the manipulation should be somehow invertible. By using the reverse instructions from the examples, we expect the model to reconstruct the original input image from the manipulated one. A high image fidelity score indicates that the model successfully preserves the instruction-invariant content of the input image during the manipulation process.
>
> In summary, the combination of query and prompts in the evaluation metric aims to assess the model's performance in capturing the underlying human intent from the visual instructions and applying the desired operations to a new image, while preserving the instruction-invariant content of the input image. We believe that this combination of directions provides a comprehensive evaluation of our model's capabilities in the context of exemplar-based image manipulation tasks. We will clarify this part in the manuscript and tune down its claim.
>
> **_Equations 4 to 6 are unclear._**
>
> We apologize for the lack of clarity in Equations 4 to 6. This section introduces the "Residual Block - Self Attention - Cross Attention" structure in diffusion model's UNet denoise predictor. In the original presentation, we used an excessive number of overlapping notations, making the description vague and unclear. We will rewrite this part to provide a clearer explanation. Thank you for your valuable feedback.
>
> **_Supplementary material is not formatted as an appendix._**
>
> We apologize for the formatting issue in the supplementary material. We will ensure that the supplementary material is properly formatted as an appendix in the revised version of the paper.
>
> **_Only injecting context/prompts through cross attention?_**
>
> It will not work effectively, because relying solely on cross-attention makes it challenging to preserve low-level information, which in turn reduces the quality of image editing. Low-level information is crucial, as many tasks require direct reference from the query image $\textbf{I}$ or instruction images $\textbf{E}$ and $\textbf{E}'$, such as replicating the unique shapes drawn by users in the instructions.
>
> Our primary motivation for constructing the 4-quadrant inpainting task is to inject low-level features from $\textbf{E}$, $\textbf{E}'$, and $\textbf{I}$ through the self-attention branch rather than relying solely on cross-attention. If we do not construct this inpainting task and solely incorporate features into the UNet's cross-attention to control diffusion, the image features would lose a significant amount of detail after being encoded by CLIP. Moreover, since the injection takes place in the latent space of LDMs, low-level information must be restored using the knowledge of the VAE's encoder.
>
> By using the grid-like image inpainting approach, features of $\textbf{E}$, $\textbf{E}'$, and $\textbf{I}$ are fully aligned with the predicted image $\textbf{I}'$, as they all pass through the VAE's encoder. This method allows us to retain as much low-level information as possible in a simple and effective manner. If the loss of resolution is a concern, we can adopt a channel concatenation approach (however, this would require allocating separate parameters for each image to avoid alignment issues).

---

> > ### Comment · Reviewer_RQEf · 2023-08-20
> > **Thank you for the response!**
> >
> > Thank you for the response! You have addressed my concerns/question and given the rebuttals to other the reviewers, I will raise my score.

---

### Author Rebuttal · Authors · 2023-08-09

We express our gratitude to the reviewers for their valuable insights and suggestions. In this section, we address a common concern raised by the reviewers regarding the task formulation.

In this study, we introduce a novel protocol for image manipulation based on visual instructions. As indicated by our title, our objective is to offer users a rapid and efficient tool for implementing desired edits. This tool operates akin to an ImageBrush, streamlining the process of customizing batches of images. This paradigm holds promising potential for various innovative applications. For instance, photographers can seamlessly apply Photoshop-style retouching to entire collections of similar images, enhancing their workflow. Additionally, users who encounter compelling editing instances from peers or pre-trained models can effortlessly apply these modifications to their own images, eliminating the need for original intricate intermediate steps or specific parameter adjustments.

According to the reviewers feedback, the major concern is that finding the intended visual instruction examples might be challenging. ​​However, it's important to note that this challenge is not insurmountable. On the one hand, many paired instruction examples could be obtained through direct manipulation using software like Photoshop. On the other hand, our model exhibits a degree of flexibility in accommodating variations between query images and the provided instruction examples, as evidenced in Figures 2 and 3 within the attached PDF document.

---

### Decision · Program_Chairs · 2023-09-21

**Decision:**

Accept (poster)

**Comment:**

The paper received overwhelmingly positive reviews, with the majority of reviewers recommending acceptance.
The main negative points raised in the reviews were in connection with missing comparison with specific previous methods and additional metrics, which were then included after the rebuttals in a way that most reviewers indicated as satisfactory.
Other outstanding doubts have to do with some constraints of visual prompting and editing workflows that use this technique more than weaknesses of this work specifically, and seems to also have been satisfactorily addressed by the rebuttals.
Reviewers praised the innovation of this work, which includes the introduction of a novel evaluation metric for exemplar-based image manipulation that does require human evaluation, the quality of the results of the empirical downstream evaluations and comparisons with alternative methods where feasible, and the exemplar-based visual prompting that it opens up.